# Foraging animals use dynamic Bayesian updating to model meta-uncertainty in environment representations

**James Webb** [iD] [1,2]*, **Paul Steffan**[1], **Benjamin Y. Hayden**[3], **Daeyeol Lee**[4], **Caleb Kemere**[1,5]☯*, **Matthew McGinley**[1,2,5]☯*

**1** Department of Neuroscience, Baylor College of Medicine, Houston, Texas, United States of America, **2** Jan and Dan Duncan Neurological Research Institute, Texas Children's Hospital, Houston, Texas, United States of America, **3** Department of Neurosurgery, Baylor College of Medicine, Houston, Texas, United States of America, **4** The Zanvyl Krieger Mind/Brain Institute, The Solomon H Snyder Department of Neuroscience, Department of Psychological and Brain Sciences, Kavli Neuroscience Discovery Institute, Johns Hopkins University, Baltimore, Maryland, United States of America, **5** Department of Electrical and Computer Engineering, Rice University, Houston, Texas, United States of America

☯ These authors contributed equally to this work.

* james.webb@bcm.edu (JW); caleb.kemere@rice.edu (CK); matthew.mcginley@bcm.edu (MM)

**Data availability statement:** Behavioral data for this study is published on Zenodo for the freely

## Abstract

Foraging theory predicts animal behavior in many contexts. In patch-based foraging behaviors, the marginal value theorem (MVT) gives the optimal strategy for deterministic environments whose parameters are fully known to the forager. In natural settings, environmental parameters exhibit variability and are only partially known to the animal based on its experience, creating uncertainty. Models of uncertainty in foraging are well established. However, natural environments also exhibit unpredicted changes in their statistics. As a result, animals must ascertain whether the currently observed quality of the environment is consistent with their internal models, or whether something has changed, creating meta-uncertainty. Behavioral strategies for optimizing foraging behavior under meta-uncertainty, and their neural underpinnings, are largely unknown. Here, we developed a novel behavioral task and computational framework for studying patch-leaving decisions in head-fixed and freely moving mice in conditions of meta-uncertainty. We stochastically varied between-patch travel time, as well as within-patch reward depletion rate. We find that, when uncertainty is minimal, mice adopt patch residence times in a manner consistent with the MVT and not explainable by simple ethologically motivated heuristic strategies. However, behavior in highly variable environments was best explained by modeling both first- and second-order uncertainty in environmental parameters, wherein local variability and global statistics are captured by a Bayesian estimator and dynamic prior, respectively. Thus, mice forage under meta-uncertainty by employing a hierarchical Bayesian strategy, which is essential for efficiently foraging in volatile environments. The results provide a foundation for understanding the neural basis of decision-making that exhibits naturalistic meta-uncertainty.

moving
(https://doi.org/10.5281/zenodo.15104572) and
head-fixed
(https://doi.org/10.5281/zenodo.15104600,
https://doi.org/10.5281/zenodo.15179590)
tasks. GitHub repositories are publicly available
for code related to data preprocessing and
analysis
(https://github.com/jbwebb8/foraging-analysis),
experimental logic and data acquisition
(https://github.com/kemerelab/TreadmillIO),
and the capacitive lick sensor
(https://github.com/kemerelab/cap-sensor).

**Funding:** Research reported in this publication
was supported by the National Institute on
Deafness and Other Communication Disorders
of the National Institutes of Health under award
number R01 DC017797 (to MM); the National
Institute of Neurological Disorders and Stroke
of the National Institutes of Health under award
number R01 NS115233 (to CK); the Integrative
Graduate Education and Training fellowship
within the Division of Graduate Education from
the National Science Foundation under award
number 1250104 (to JW); and financial support
from the Dunn Foundation (to MM). The
funders had no role in study design, data
collection and analysis, decision to publish, or
preparation of the manuscript.

**Competing interests:** The authors have
declared that no competing interests exist.

## Author summary

The ethological approach to understanding how animals make decisions is to use tasks that they often face in their natural environments. One such task, canonically termed patch-based foraging in behavioral ecology, involves harvesting resources from spatially separated areas (termed "patches") that deplete over time. While patch foraging, animals must choose when to leave each patch to find a new, replenished one. The marginal value theorem (MVT), describes the optimal behavioral strategy when environment statistics are stable and known to the animal. However, naturalistic settings are often noisy and uncertain, which limits the applicability of the MVT. Here, to understand how laboratory mice make ethologically relevant decisions, we implemented a patch-based foraging task in either a physical or virtual patch-based foraging environment. The tasks incorporate uncertainty in the richness of patches, the distance between patches, and, importantly, the randomness of reward timings within a patch. When randomness of reward timing was low, animals behaved in a manner consistent with the MVT. However, when reward-timing randomness was high, mice dynamically weighted average statistics and recent observations, captured in a Bayesian estimator. Our results thus lay the groundwork for studying how the brain solves tasks when presented with multiple levels of uncertainty.

## Introduction

When foraging, the optimal policy should maximize reward rate (rewards per unit time) [1]. In the most well-studied class of foraging decisions, an animal within a patch of resources needs to decide when to abandon the depleting patch and pay a cost (normally in the form of a travel time) to move to a newer, richer one [1–3]. There has been increasing interest in foraging behavior as a potential avenue to understand normal and aberrant decision-making and, in animals, as a tool for mechanistic understanding of the neural circuit basis of decision-making [4,5]. This interest is reflected in extensive research in ethology and behavioral ecology [6–12]. Furthermore, deviations from optimal foraging are diagnostic of learning deficits and psychiatric illnesses [13,14].

When the environmental parameters are completely known to a forager, optimal behavior is dictated by the marginal value theorem (MVT), which shows that leaving times are reward-rate maximizing when marginal reward declines to match the average of the environment. However, the MVT makes the rather strict and unrealistic assumptions that the environmental statistics are stationary and that the forager has a perfect internal model of those statistics. If the forager does not have a good environmental model, each outcome they face poses a challenge. Does the outcome fit with, or deviate from, their internal model? This challenge arises from uncertainty about the environment resulting from its stochasticity and nested meta-uncertainty about whether those environment statistics have undergone a change. For example, consider an apple encountered under a tree or at a supermarket. If that apple has an unusual color or below-average sweetness, the forager must decide whether it is part of the normal variability in quality apples or instead if they should move to a new tree or store. As a result of this meta-uncertainty, decision-makers must constantly evaluate whether the variability they encounter reflects stochasticity in known environmental statistics or a change in those statistics. Both forms of variability are naturalistic, as evidenced in the ecological literature [15–18], and consequently should be accounted for in strategy.

As natural foragers, rodents often encounter such meta-decisions in their native environments, needing simultaneously to parse the economics and risks (e.g. predation), as well as their variability on multiple time scales [19–22]. Given their predilection for such tasks, and the wide use of rodents in systems neuroscience, recent laboratory studies have utilized foraging constructs to explore behavioral strategies and their underlying neurophysiological mechanisms [23–30]. However, replicating the dynamics of natural foraging in an experimental setting is difficult. Within patches, reward encounters should contain some level of variability while also exhibiting sufficient stability on which rodents can base patch-leaving decisions. Additionally, the environmental information contained in the encounters should be perceptible to the animal and lead to interpretable outcomes. Striking a balance between replicating the naturalism needed to tap into rodents' innate cognitive abilities, while creating experimental constructs for which meaningful behavioral and/or neurophysiological data can readily be acquired, analyzed, and interpreted, creates a dilemma for the experimenter [31]. In particular, meta-uncertainty has not been accounted for in laboratory foraging behavioral models.

Here, we implemented a patch-based foraging task in freely moving and head-fixed mice that captured several key naturalistic elements, including meta-uncertainty. Mice run between reward ports on a physical or virtual track and receive liquid rewards at a rate that decays within a patch over time. Patch location, and richness, are indicated with acoustic cues. At any moment within a patch, mice can leave and travel to the next one, which is replenished to its initial reward rate. Critically, we introduced stochasticity in the depletion process, so that sequences of reward encounters contain both informative and stochastic elements, confronting mice with the need to parse whether deviations in reward timing reflect the stochasticity in environmental parameters or their uncertainty about the environment. Our results show that a hierarchical model in which outcome variability is separated from environmental stability by a Bayesian estimator better explained behavior than simple heuristics or standard MVT models. Consequently, our study illuminates how mice adapt to a ubiquitous phenomenon in their natural environment – meta-uncertainty – through calculated behavioral strategies, underscoring a previously unknown layer of complexity to decision-making that supports robust behavior in the presence of environmental volatility.

## Results

### A patch-based foraging task in freely moving mice

To study patch-based foraging behavior under uncertainty in the laboratory, we developed a freely moving mouse preparation. We constructed a linear track system, similar to designs typically used to study hippocampal function during navigation [32,33]. After undergoing a two-step training regimen to become familiar with the experimental setup (see Methods and materials), mice successfully performed the freely moving patch-based foraging task.

The task consisted of running back and forth between either end of the linear track where reward ports dispensed a stochastically depleting sugar-water resource, with acoustic cues for reward availability (Fig 1A, top right). At the start of each session of the task, a mouse was placed in the center of the linear track. Upon navigating to either of the two reward ports, a tone cloud was played continuously from a speaker near the reward port, indicating that rewards were available (Fig 1A, bottom left). Upon nose-poking within the reward port, mice could lick a spout for liquid rewards. We refer to this nose-poked licking behavior as harvesting, in line with behavioral ecology literature [1]. Reward availability followed an inhomogeneous gamma process (IGP; also known as a modified inhomogeneous Poisson process), with an exponentially decaying Poisson rate (Fig 1B, bottom left). Pure tones played at the time of

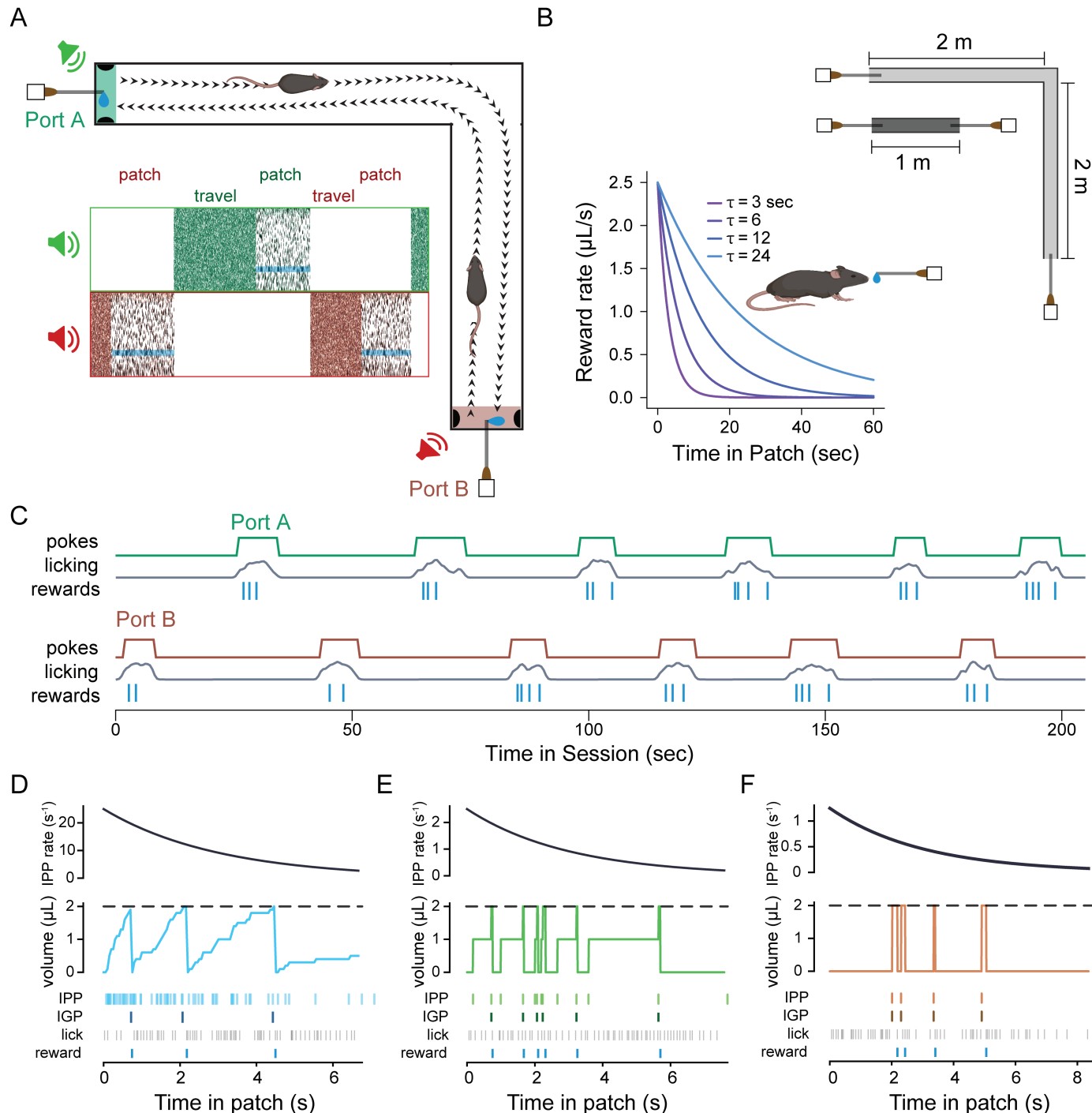

**Fig 1. A patch-based foraging task for mice on a linear track.** (**A**) Top right, schematic of freely moving foraging task, showing that animals navigate between two reward ports at either end of a track; by nose-poking into port A or B, animals can receive sucrose solution rewards. Bottom left, spectrogram of sounds from the two speakers; while poked into a port (patch), a tone cloud stimulus played through an adjacent speaker, with intermittent pure tones indicating reward availability (opaque cyan bands); after leaving one port, and while traveling to the other (travel), a pink acoustic noise is played at the opposite speaker until the animal pokes into the adjacent port. (**B**) Environmental perturbations include two track lengths (illustrated at top) and four time constants for the exponentially decaying reward rate (illustrated at bottom). (**C**) Example data from twelve consecutive patches during the first 2.5 minutes of a session on the 1 m track with reward decay rate of 3 seconds. Traces of the digital poke signal, smoothed lick rate, and a raster plot of reward times, are shown for the two reward zones (green indicates Port A and red indicates Port B, following the color scheme in **A**). (**D-F**) The reward-generating process is shown for an example patch in environments with a low (**D**; *RSI* = 0.05; light blue), moderate (**E**; *RSI* = 0.5; green), or high (**F**; *RSI* = 1.0; brown) level of variability in reward timing. Top: Black curves indicate

the inhomogeneous Poisson process (IPP) with an exponentially decaying rate, which generated events in a patch. Middle: Colored stairstep curves indicate the time and volume associated with each event, $V_0$, added to a potential-reward reservoir that accrued over time. Once the volume of the reservoir reached a threshold (dashed line; 2 $\mu L$ for all environments), 2 $\mu L$ of sucrose solution reward became available for the animal to receive upon licking. Once the reward droplet was given, the reservoir was depleted by the reward volume. Note that volume continued to accrue in the reservoir even after the threshold had been reached. Bottom: Light colored rasters indicate the time of each addition to the reward reservoir from the IPP; dark colored rasters indicate the times of available reward (IGP); lick times are indicated with grey rasters; received-reward times are indicated at very bottom in blue. All examples are shown for environments with $\tau$ = 3 seconds.

reward availability provided information about reward timing that was independent of licking behavior. Because the underlying rate for the IGP exponentially decayed over time, availability of rewards became increasingly rare as the animal remained in the patch, simulating classical patch depletion in behavioral ecology [1]. The level of stochasticity in reward dynamics was varied between three levels and was quantified using a reward stochasticity index (RSI), defined as the ratio of the hidden event volume to observable reward volume (see Methods and materials). A larger RSI value corresponded to greater variance in the timing of rewards, independent from the decay rate.

While mice were engaged in a nose-poke at a reward port, they could terminate harvesting by un-poking, at which point the tone-cloud stimulus stopped playing to indicate the port was inactive. Un-poking immediately triggered acoustic pink noise to play from a speaker near the opposite port, cueing the mouse that reward was available at that port. Upon traveling to and poking in the opposite port, the auditory cue switched to the tone cloud, and mice could receive rewards from the depleting IGP, as previously. The travel distance, and thus the opportunity cost imposed by lost time by traveling between ports, was varied systematically and unambiguously by using two tracks with different lengths (Fig 1B, top right). A single set of fixed environmental parameters was used for each behavioral session.

## Mice adapt their behavior to daily perturbations in the environmental statistics

A cohort of mice ($N$ = 8 mice, 27.6 $\pm$ 1.1 sessions per animal) were run in the patch-based foraging task, at the low stochasticity level. For each session (one per day), the reward decay rate had one of four values ($\tau$; 3, 6, 12, or 24 seconds; Fig 1B, bottom left) and used one of two track lengths (1 meter or 4 meters; Fig 1B, top right). Across environments, mice learned to alternate between reward ports and lick for rewards (Fig 1C), encountering a substantial number of patches per session (normal distribution; $\mu$ = 45.58, $\sigma$ = 22.68) and remaining in patches for a wide range of poke durations (log-normal distribution; $\mu_{log_{10}}$ = 0.91, $\sigma_{log_{10}}$ = 0.29), termed the patch-residence time [1].

We defined residence time as the time from reward port entry (via poking) to exit (via un-poking) and non-harvest time as the time between exit from a reward port to entry at the next active reward port. Because animals also exhibited non-foraging behaviors, such as exploring or grooming, we estimated the task-relevant non-harvest time (referred to as 'travel time') as the tenth percentile of all durations of time between patches, for each animal, on each track type, although results were robust across a wide range of percentile values (S7 Fig). Both the full non-harvest (Fig 2A) and task-relevant travel time estimates (Fig 2B) indicated that track length affected the temporal cost of traveling between reward ports. To understand the combined influence of reward depletion rate and travel time on harvesting behavior, we tested their effect on residence time using a cluster bootstrap design (S5 Fig; see Methods).

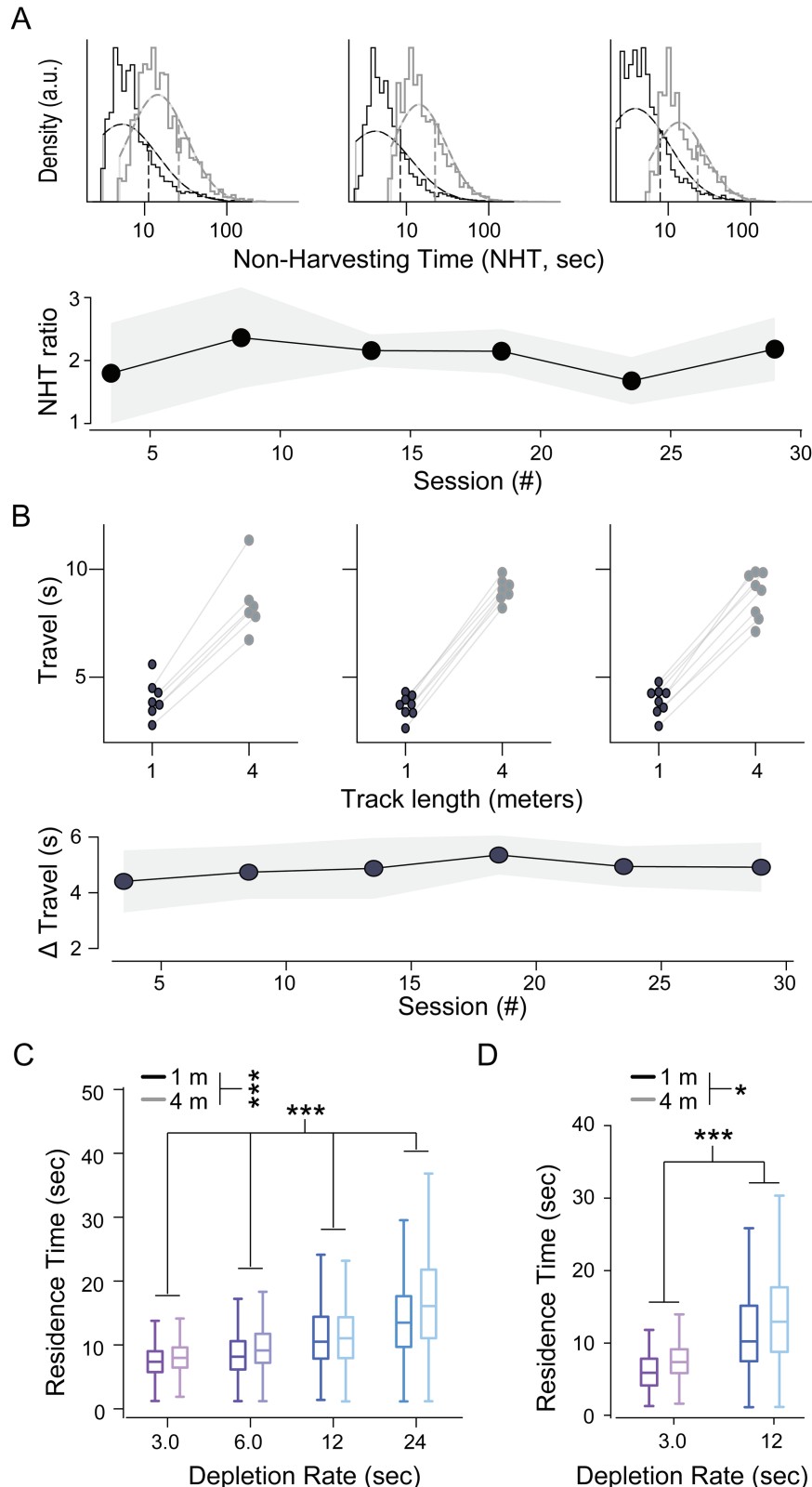

**Fig 2. Mice adapt their patch-residence time to within- and between-patch environment statistics.** (**A**) Comparison of total non-harvesting times between track types. At top, histograms (solid lines) and fit of log-normal distributions (dashed curves) for all total travel times in five-day bins for early (left), middle (center), and late (right)

in training on the 1 m (black) and 4 m (grey) tracks. At bottom, non-harvest time (NHT) followed a log-normal distribution and therefore was compared between short and long tracks using the ratio of the geometric mean for the 4 m track to the geometric mean for the 2 m track. Each point is centered on the five-day, non-overlapping bin for which the ratio was calculated. (**B**) Comparison of task-relevant travel times between track types. At top, examples from five-day bins in early (left), middle (center), and late (right) training are shown. Task-relevant travel times for each animal on the 1 m (black dot) or 4 m (grey dot) track are connected by light gray lines. At bottom, the mean difference between the two task-relevant travel times for each animal are binned and plotted as in **A**. Line and shaded area in **A** and **B** are the average value and standard deviation across animals. (**C**) Box plots of residence times for the low-stochasticity sessions. Results are stratified by reward decay rate (indicated on x-axis and with color) and track length (dark, 1 m track; light, 4 m track). Boxes represent the interquartile range (IQR) of residence times from all animals in the given environment. Whiskers extend 1.5x the IQR from the box edges. Center lines represent the median. (∗, p < 0.05; ∗ ∗ ∗, p < 0.0001; cluster bootstrap analysis) (**D**) Same as in **C**, but for data pooled across the high- and moderate-stochasticity environments.

We found that both parameters affected residence times and that the direction of the behavioral adaptations were in agreement with the MVT (Fig 2C; decay rate: $r = 0.50\,[0.46, 0.55]$ (mean [95% CI]), $p(r > 0) > 0.9999$; track length: $r = 0.10\,[0.05, 0.16]$, $p(r > 0) > 0.9999$).

We noticed that patch residence time decreased gradually over the course of a session, a time-on-task effect possibly resulting from fatigue or satiety [34,35]. We also noticed substantial variability between animals in overall residence times. To quantify how both experimental manipulations (decay rate, travel distance) and these confounding factors (time-on-task and individual-specific bias) affect behavior in a single model, we fit a linear mixed model (LMM) to the dataset. We set the reward decay rate, task-relevant travel time, and time-on-task as fixed effects and animal identity as a random effect. In the low-stochasticity environments, the effects of decay rate and travel time were highly significant and consistent with MVT; slower decay rates and longer travel times were associated with increased residence times (Table A in S1 Table).

## MVT-based models outperform simple heuristics at explaining the behavior

Although normative models of patch-based foraging are governed by the marginal value theorem (MVT), and the above analysis showed that our mouse results were consistent with the major predictions of the MVT, animals often solve tasks by applying simple heuristic decision-making strategies [16,36–38]. Utilizing simple heuristics may reduce cognitive demand while achieving adequate reward rate for survival needs. On the other hand, using MVT-based models would maximize the rate of reward [6,39]. Therefore, before proceeding to more complex models, including those related to reward stochasticity, we evaluated whether the strategy the mice took in their foraging behavior followed a simple heuristic that approximated MVT-based behavior, by fitting predictive models to animals' residence times.

We identified three 'local' (within-patch) heuristic rules that the mice may be employing. Namely, animals may leave a patch: (1) after a fixed duration since patch entry [heuristic, constant time; HEU-CT39], (2) after a fixed number of encountered rewards [heuristic, number of rewards; HEU-NR40], or (3) after a fixed amount of time since the previous reward [heuristic, elapsed time since reward; HEU-ETR6]. For each animal, we used the mean of each relevant metric (i.e. the average duration, number of rewards, or delay between last reward and patch-leaving per patch) to predict the residence time in each patch (Fig 3A–3B). We first applied these models to the low-stochasticity regime.

We compared results of the heuristic models to two MVT-based models (Fig 3C). In the first MVT model, we predicted the residence time for each environment by optimizing the overall reward rate given knowledge of the underlying parameters, which equates to the optimal residence time in classic foraging theory (MVT, optimal; MVT-OPT). Because animals may generate stable but inaccurate internal estimates of the task parameters, in the second

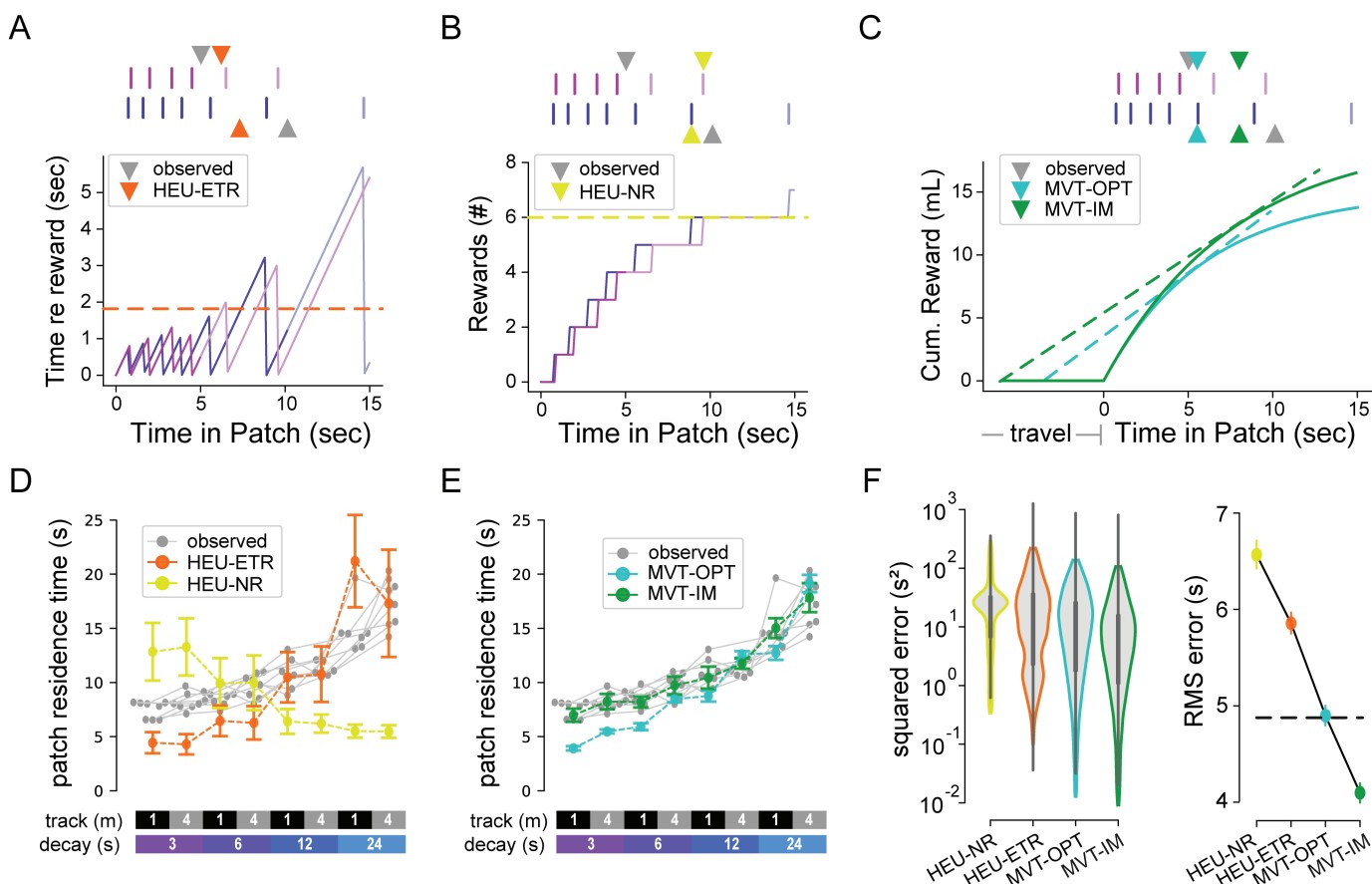

**Fig 3. Foraging behavior is better explained by MVT models than by simple heuristics.** (**A**) Schematic illustrating the heuristic model of patch-leaving based on elapsed time without reward (HEU-ETR). At top, reward sequences from two example patches (purple and blue vertical lines) are displayed from a session on the 1 m track with reward decay $\tau$ of 6 seconds. Grey triangles indicate the observed patch leaving (residence) time for each patch. Orange triangles indicate the predicted time of patch leaving from the HEU-ETR model for the same reward sequences. At bottom, purple or blue traces represent the current time since the last reward in the given patch (dark purple or blue) and a simulation of the expected (light purple or blue) time if the animal had not left the patch. Once the threshold criterion is exceeded (dotted orange line), the model predicts patch-leaving. (**B**) Schematic illustrating the heuristic model of patch-leaving based on the number of received rewards in the patch (HEU-NR). At top, same as **A**, except that mustard triangles indicate the predicted time of patch leaving from the HEU-NR model for the associated reward sequences. At bottom, example traces similar to **A**, but displaying the number of observed rewards as a function of time in patch. Note the need for computing expected future reward times for the first (purple) example patch. (**C**) Schematic illustrating the two models of patch-leaving based on the MVT. At top, same as **A**, except that the cyan triangles indicate the predicted time of patch-leaving from the optimal MVT model (MVT-OPT) and the green triangles for the predicted time for the internally modeled MVT model (MVT-IM), for the associated reward sequences. Both MVT models learn the travel time between patches (horizontal line at beginning of the cyan and green curves) and the average reward function (integration of decaying exponential of the cyan and green curves). Patch-leaving occurs when the marginal rate (dashed tangent lines) equals the average rate for the overall environment (tangent line extrapolated to beginning of travel). (**D**) The per-animal average observed patch residence time (grey dots and lines) for each track length and reward condition (indicated at bottom). Mustard and orange dots and dashed lines indicate the model predictions from the two heuristic models (HEU-NR and HEU-ETR, respectively). (**E**) Same as in **D**, but for the two MVT models. For **D** and **E**, colored lines and error bars represent the mean and standard deviation, respectively, of the model predictions for patches pooled across all animals in a given environment. (**F**) At left, the model prediction error. Black vertical lines represent the 95% confidence intervals, which were bootstrapped from the set of prediction errors. At right, the root-mean-square error (RMSE) of each model. The null model (HEU-CT) RMSE, which is equivalent to the average standard deviation of residence times across animals, is shown as the black dashed line.

MVT model, the predicted residence times for each animal followed the MVT, but with environment-specific parameter estimates that were fixed for each environment, but not necessarily correct (MVT, internal model-based; MVT-IM). This model assumes that animals attempt to maximize their overall harvest rates following the MVT, based on model parameters that reflect their perceived, or internally estimated, reward decay rates and travel times

for each environment. In doing so, the environmental parameters in the MVT-OPT equation were replaced by those values that best predicted the empirical residence times (S6 Fig).

When comparing the heuristic and MVT models, we used the fixed duration heuristic model (HEU-CT) as a null hypothesis. The other two heuristic models had poor fits to the observed data, both qualitatively and quantitatively. The HEU-NR model predicted an opposite trend for the dependence of patch residence time on decay rate to what was observed in the data (Fig 3D, mustard yellow). The HEU-ETR data underestimated residence time for fast reward decay rates and, by construction, could not capture the effects of the track length (Fig 3D, orange). The MVT-OPT model qualitatively captured both the effects of track length and reward decay but, like the HEU-ETR, underestimated patch residence time for fast decay rates (Fig 3E, cyan). Relative to the optimal time according to MVT, animals remained too long in patches, or overharvested, particularly in environments with fast decay rates (Fig 3E) as has been observed previously [41,42]. The MVT-IM captured the data well, with no systematic errors (Fig 3E, green), including accounting for overharvesting in fast-decay rate environments. Thus, animals adapted to environmental perturbation in agreement with the MVT, but as if they underestimated the patch reward decay rate (Fig 3F and S6; root-mean-square prediction error (RMSE) [95% CI]: HEU-CT, 4.88 [4.75, 5.01]; HEU-ETR, 5.86 [5.75, 5.97]; HEU-NR, 6.57 [6.42, 6.71]; MVT-OPT, 4.90 [4.80, 5.00]; MVT-IM, 4.10 [4.00, 4.20]).

## Local reward sequences dynamically influence patch-leaving decisions

The mice exhibited substantial variability in patch-leaving time within each session. We hypothesized that this within-session behavioral variability may stem from two sources of uncertainty: (1) animals do not know the daily patch decay parameters, and (2) there is patch-to-patch stochasticity in reward availability. We thus sought to determine whether recent reward statistics influence their choices. To do so, we implemented a Bayesian model with knowledge of the underlying Poisson process to generate a maximum likelihood estimate (MLE) of the current reward rate given a set of observed reward times. We limited the model input to reward times in the current patch and calculated the MLE and true Poisson reward rates at patch-leaving. We then compared the error of the Bayesian model estimate at patch-leaving to the deviation of the current residence time from the average of all residence times in a given session.

If animals tracked the immediate reward rate to determine the leaving time, as proposed by MVT, then overestimating the reward rate would lead to longer residence times (Fig 4A, left) and vice versa (Fig 4A, right). Linear regression showed a significant positive correlation between the Bayesian rate estimation error and the deviation of residence time from average, in the low stochasticity environment (Fig 4B; observed: $r = 0.52 \pm 0.01$, mean $\pm$ standard deviation across five-fold cross-validation subsets; $R^2 = 0.27 \pm 0.01$; shuffled: $r = 0.18 \pm 0.01$, $R^2 = 0.032 \pm 0.002$). Thus, the animals' leaving times are influenced by the local (current patch) reward sequence information, even when reward stochasticity is low.

## Animals utilize both local and global information in highly stochastic environments

We next tested whether the behavioral strategies observed in the low stochasticity environments extend to more highly stochastic environments. We ran the animals that had previously performed the freely moving foraging task with $RSI = 0.05$ in the same task, except with increased variability in reward timing ($RSI \in [1.0, 2.0]$; $N = 8$ mice, $10 \pm 1$ sessions per animal). The task was structured such that average reward dynamics remained unchanged, but the variance of reward timing between patch encounters increased substantially (Fig 1E–1F).

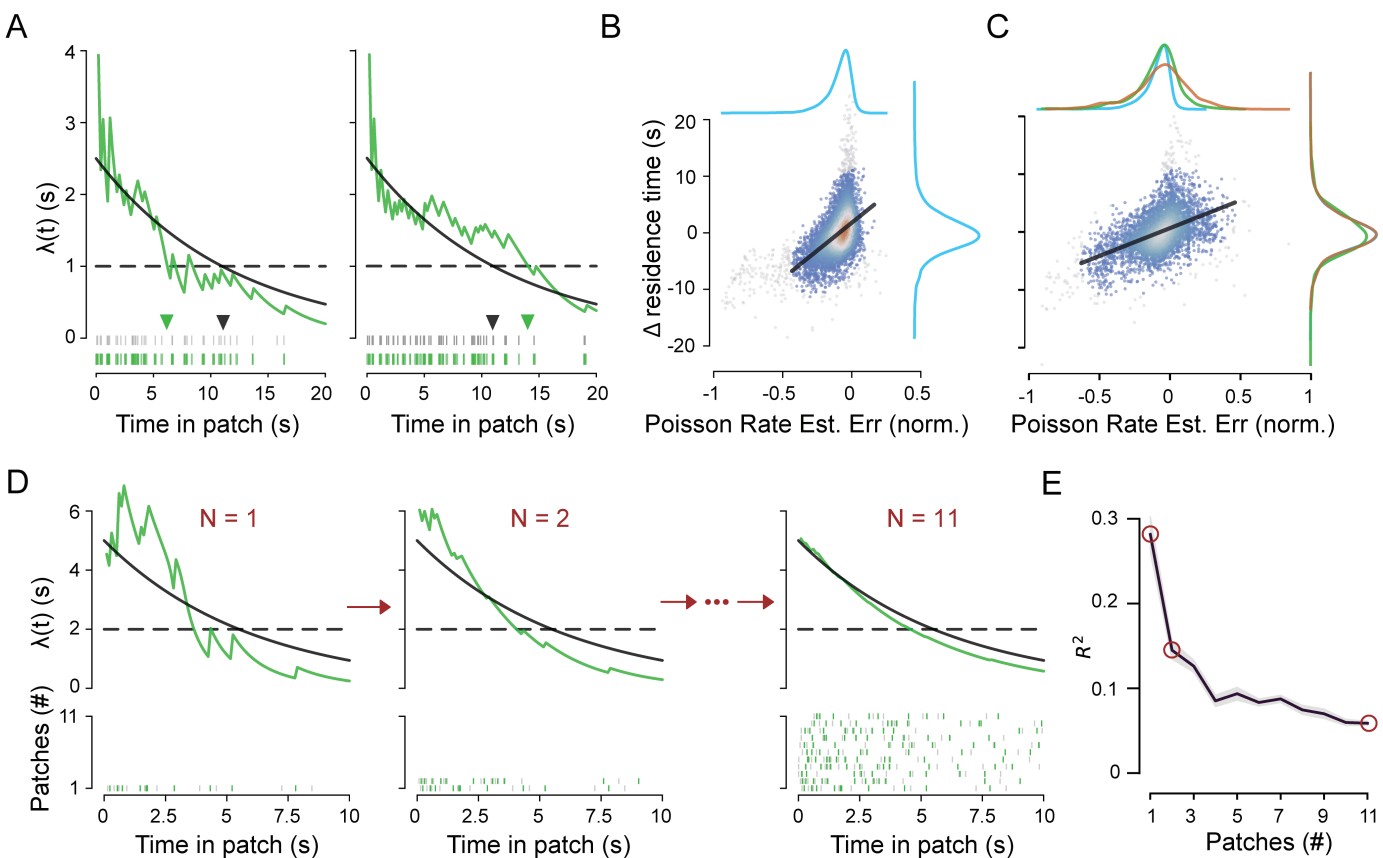

**Fig 4. Mice dynamically adjust patch-leaving time based on recent patch reward sequences.** (**A**) Two example time-varying Poisson reward rates in the moderate stochasticity context. The maximum likelihood estimate (MLE, green curves) of the true underlying Poisson rate (black curve) was calculated at 500 ms intervals for two example sequences generated from an inhomogeneous gamma process (IGP). Reward times are shown in the rasters at bottom, with grey and green vertical lines representing event and reward times, respectively. Triangles indicate the optimal leaving time based on the true reward rate (black triangle) or MLE of the reward rate (light green triangle), respectively, which occurs when the immediate reward rate falls below the average reward rate in the environment, indicated with dotted horizontal black line. The example sequences were generated from an environment with *RSI* = 0.5 for illustration purposes. (**B**) For each patch encounter in the *RSI* = 0.05 environments, the difference between the true and the maximum likelihood estimate (MLE) of the Poisson rate at patch leaving (plotted on the x-axis) is scattered against the deviation of the current residence time from the session average (y-axis). Color represents the neighboring density of points, ranging from low (blue) to high (red). The regression line (black) was fit to points within the 99% confidence ellipse from a bivariate Gaussian distribution fit to the data. Points lying outside of the confidence ellipse are colored gray. Kernel density estimates of the marginal distributions (Gaussian kernels; bandwidth estimated via Scott's rule) are shown in the margins. (**C**) Same as in **B**, but for the moderate (green marginal) and high (brown marginal) stochasticity environments. The low stochasticity marginals (blue) are also shown, for visual comparison purposes. Moderate and high stochasticity data are pooled for the scatter plot and regression fit. (**D**) The MLE of the Poisson rate is shown for a given sequence of rewards in a patch using different degrees of observation history to generate the estimate. At top, the MLE of the Poisson rate (green) was computed using observed rewards in the current patch only (*N* = 1, left) or in addition to observed rewards in the previous one (*N* = 2, middle) or ten (*N* = 11, right) patches. The true Poisson rate (solid black) and example leaving threshold (dotted black) are also shown. At bottom, raster plots display the IPP events (gray) and IGP observations (green) used to generate the MLE above. Each row represents sequences in one patch, with the oldest being at top (patch 11) and the current patch, for which all three MLEs are computed, at bottom (patch 1). (**E**) For pooled data from the moderate and high stochasticity environments, the coefficient of determination for the change in residence time vs. the rate estimation error was computed as in **B** and **C** using various degrees of observation history in the MLE of the Poisson rate at patch-leaving. The mean (solid line) and standard deviation (shaded area) of the five cross-validation subsets are shown. As in **D**, *N* represents the total number of recent (current plus prior) patches included in the MLE calculation. The red circles correspond to the examples from **D**.

In high-stochasticity environments, mice still shifted residence times with decay rate and track length in accordance with MVT (Fig 2D). Cluster bootstrap analysis showed these changes to be statistically significant (decay rate: $r = 0.50\,[0.46, 0.55]$ (mean [95% CI]), $p(r > 0) > 0.9999$; track length: $r = 0.10\,[0.01, 0.18]$, $p(r > 0) = 0.985$). To further test behavioral

adaptations, as above, we fit the behavioral data in the high-stochasticity environments with a LMM using the same explanatory variables as the LMM fit to the low-stochasticity data. The model showed significant adaptations in residence times, in the directions expected for the MVT, for both decay rate and track length manipulations (Table A in S1 Table).

We then assessed behavioral strategies by fitting the same local heuristic and MVT-based models to residence times in high-stochasticity environments (Fig 5B–5C, left). Of note, models based on average reward dynamics, including all the local heuristic and MVT models, made predictions that did not depend on RSI and consequently predicted similar residence times to those in the low-stochasticity environments. Consistent with low-stochasticity environments, the models based on the number of observed rewards (HEU-NR) and perceived MVT parameters (MVT-IM) were the worst- and best-performing, respectively (RMSE [95% CI]: HEU-CT, 4.92 [4.72, 5.12]; HEU-ETR, 4.67 [4.48, 4.92]; HEU-NR, 7.03 [6.82, 7.24]; MVT-OPT, 4.85 [4.67, 5.05]; MVT-IM, 3.98 [3.81, 4.16]), suggesting that animals effectively extracted average dynamics from stochastic observations. However, in contrast to environments with low stochasticity, in high stochasticity, the elapsed time without an observed reward heuristic model partially explained variance in residence time, implying that animals defaulted to tracking this simple metric when reward timing was more variable. Nevertheless, despite more unpredictable reward sequences, animals demonstrated behavioral adaptations consistent with the MVT.

We next leveraged these sessions to further explore whether animals were making continuous, dynamic estimates of reward parameters, as was indicated by models of the low-stochasticity environment sessions (Fig 4B). As before, we computed the MLE of the Poisson rate at patch-leaving using the current reward sequence and compared it with the change in residence time relative to the session average (Fig 4C). The correlation was both positive and significant ($RSI \in [0.5, 1.0]$; observed: $r = 0.49 \pm 0.02$ (mean $\pm$ standard deviation across five-fold cross-validation subsets), $R^2 = 0.24 \pm 0.02$; shuffled: $r = 0.06 \pm 0.01$, $R^2 = 0.004 \pm 0.001$; see S2 Fig), consistent with the hypothesis that mice use recent reward history to modify the global patch-leaving decision. Interestingly, while the variance of the estimation error increased with increasing $RSI$, as expected, the variance of the residence times remained unchanged (see marginal distributions in Fig 4C). Thus, increasing the stochasticity of in-patch reward dynamics did not affect the overall within-session behavioral variability but rather coupled that decision variability to the broadened distribution of reward sequences.

## Animals use recent observations to update their estimates of environmental variables

We next explored how performance in the current patch was influenced by the recent patch history. We followed the same procedure as above to compare the error of the estimated reward rate at patch-leaving with the variation in residence time, except that the MLE of the Poisson rate incorporated reward sequences from prior patches in addition to the sequence from the current patch (Fig 4D). Notably, the correlation with local adaptations in residence time inversely correlated with the degree of recent history used to estimate the reward rate, suggesting that large variations in the patch-leaving decision resulted from the timing of the most recently observed reward sequence (Fig 4E).

While these findings demonstrate a relationship between estimated reward rates and adaptations in residence times, they do not generate actual predictions of residence times given the estimated rates. To do so, we constructed a predictive model for patch residence times that utilized the MLE of the reward rate as input. Consistent with the MVT, the model presumed that animals left patches when the immediate reward rate fell below a given model. However,

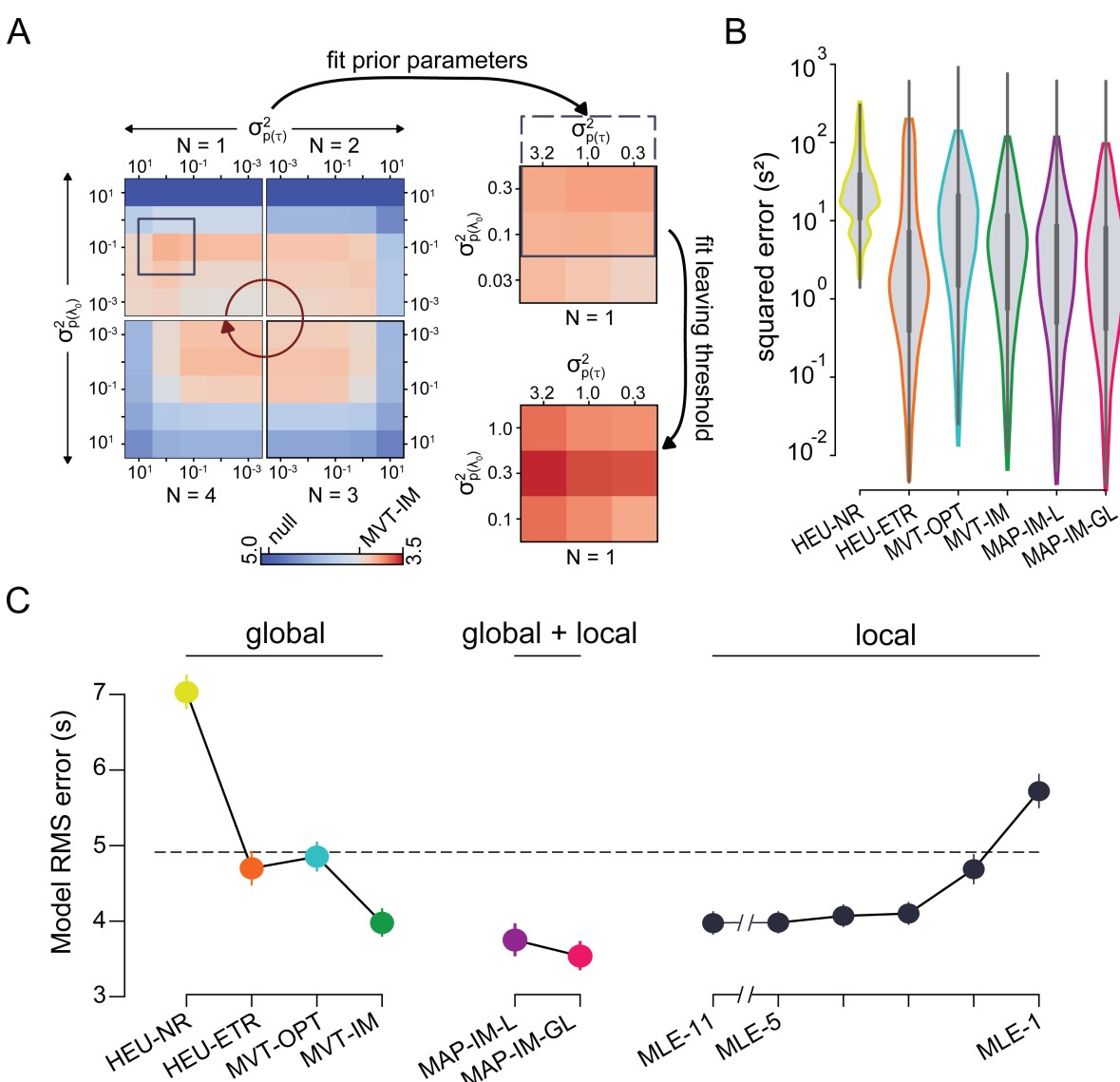

**Fig 5. Animal patch-leaving decisions reflect estimates of both the global and the local environment.** (**A**) The prediction error was calculated for models using the maximum a posteriori (MAP) estimate of the Poisson rate. At left, the root-mean-square prediction error (RMSE) is shown for MAP models with different variances of the prior probability distributions for both $\tau$ (horizontal axis) and $\lambda_0$ (vertical axis), as well as various degrees of observation history ($N$ as in Fig 4; red arrow indicates increasing order). The set of values for $N$ and variance of the priors that minimized prediction error (purple box) was analyzed at finer resolution of the prior distributions (top right). After further refining the analysis window (purple solid and dotted box), the prediction error was recomputed after additionally choosing a leaving threshold that best fit the experimental data (bottom right). The colorbar (bottom left) indicates the RMSE values and was scaled such that the RMSE of the MVT-IM model lies at its center. (**B**) The squared prediction error for the moderate and highly stochastic environments is shown for various models as in Fig 3F. (**C**) The root-mean-square prediction error for the moderate and highly stochastic environments is organized by models that used only global information (left), only local information (right), and a combination of both global and local information (middle), and is shown as the mean (dots) and standard deviation (error bars) over the five cross-validation sets. Model abbreviations are described in detail in the text. The RMSE of the null model (HEU-CT) is displayed as the black dotted line.

the new model utilized the estimated, as opposed to fixed, reward rate (MLE-x, where x represents the number of recent patches included in the likelihood estimate of the Poisson rate). As in Fig 4E, the MLE of the Poisson rate for each patch was calculated using a given degree

of patch history. The reward rate threshold corresponded to the reward rates at patch-leaving in the MVT-IM model. Model prediction error decreased with the extent of patch history, in contrast to local adaptations of residence times, and asymptotically approached the accuracy of the MVT-IM model (Figs S3A and 5C).

Capturing variability at different timescales thus led to ostensibly conflicting suggestions about the degree of patch history incorporated into patch-leaving decisions. Patch-to-patch variability in residence times was best explained by the most recent observations, whereas the mean residence time across a session was best explained by incorporating all previous observations, including distant ones. In other words, while recent experience influenced patch-to-patch variability in patch-leaving decisions, it did not provide sufficient evidence, in the form of reward rate estimation, to fully predict patch-leaving decisions.

This discordant relationship with the degree of observation history suggested a model in which local and global estimates of the environment were independently computed and made distinct contributions to behavior. To address this, we used a Bayesian approach to incorporate the prior probabilities of the underlying reward rate parameters (initial reward rate and decay rate), as well as the MLE of the reward rate, which reflected the global and local features of the environment, respectively. The resultant estimate of the reward rate, termed the *maximum a posteriori* (MAP) estimate, extends the MLE by modulating the likelihood by prior beliefs, which, in this case, reflect the mouse's perception of average statistics. The model predicted patch-leaving to occur when the MAP estimate of the reward rate fell below the reward rate threshold, which corresponded to the parameters of MVT-IM for a given environment. Based on the results from Fig 4E, we used observed reward times from the current patch and up to the previous three patch encounters.

To determine the best model parameterization, we first conducted a grid search by computing the prediction error across a range of prior distributions that were centered on the reward parameters of MVT-IM but differed in variance, reflecting the degrees of uncertainty in the global estimate (Figs 5A and S3B–S3D). Consequently, these models reflected the animal's perception of environmental parameters (internal model-based) in a probabilistic framework (MAP) that was updated by recent experience (local optimization). The best-fit model (MAP, internal model-based, local optimization; MAP-IM-L) utilized observations from only the current patch ($N = 1$) with moderate uncertainty in the reward rate parameters ($var(\lambda_0) = 0.3$, $var(\tau) = 0.3$); see Methods and materials for a description of the parameters). The improvement compared to MVT-IM was not significant (Fig 5B–5C; RMSE [95% CI]: 3.75 [3.558, 3.956]).

Although the MAP-IM-L model adopted parameters from the MVT-IM model to set the reward rate thresholds for patch-leaving decisions, the thresholds that best explained animal behavior might differ once the local adaptions were captured through the probabilistic framework. Consequently, we constructed a model in which both the parameters of the prior distributions and the reward rate thresholds were optimized with a hierarchical approach (MAP, internal model-based, global and local optimization; MAP-IM-GL). For each set of prior distributions, the reward rate thresholds that minimized prediction error were computed for each environment (S3D Fig; see Methods and materials). When assessed for accuracy in predicting residence times, the best-fit MAP-IM-GLmodel ($N = 1$, $var(\lambda_0) = 0.3$, $var(\tau) = 3.2$) significantly outperformed all other behavioral models that utilized only global or local environmental features (Fig 5B–5C; RMSE [95% CI]: 3.54 [3.37, 3.72]).

## Behavior in a head-fixed virtual foraging task follows MVT principles, not simple heuristics

Because head-fixed behavior allows for a wider range of physiological methods, we next tested whether the freely moving task could be adapted to a virtual patch-based foraging task for head-fixed mice. In the virtual foraging task, mice ran on a cylindrical treadmill in a 1D virtual space, using the same auditory cues as used in the freely moving task (Fig 6A). Patches were separated by a virtual track distance that the animal was required to traverse on the wheel in order to reach the next virtual patch (Fig 6B). Mice began in a patch at the start of the task. The acoustic tone cloud presented while the mice remained stationary, signaling that they were in a patch. Pure tones were embedded whenever reward was available, following the inhomogeneous gamma process, at which point mice received reward upon licking. As before, sucrose solution rewards were a constant volume of $2 \mu L$ and had increasingly longer intervals between them as time in patch progressed. At any time in a virtual patch, mice could begin walking or running on the treadmill, which signaled a patch-leaving decision, and pink noise began to play to indicate they were no longer in the patch. As they approached the next patch in virtual space, pink noise increased in intensity until they had covered the full virtual track distance for a given environment, at which point the sound switched to tone cloud. The reward-generating process began when the mice had additionally become stationary, which signaled recognition of patch entry. This sequence of virtual patch residence and inter-patch travel continued for the duration of the session.

After a training period to familiarize with the head-fixed apparatus, mice performed the foraging task with three reward decay rate ($\tau \in [3\,s, 6\,s, 12\,s]$) and three virtual track lengths (60 cm, 100 cm, 200 cm) in both low- ($RSI = 0.05$) and high- ($RSI \in [0.5, 1.0]$) stochasticity environments. As in the freely moving task, a single set of parameters was used for each session. Using similar criterion as the freely moving task, low-performing sessions and animals were removed from the analysis. Within the remaining sessions, task-relevant behavior was estimated both between and within patches.

We defined the task-relevant travel time as the time during which animal velocity exceeded the threshold for patch entry (0.5 cm/s), which accounted for 70% of the total travel time across all included sessions (S4D Fig). Moreover, animals ran continuously to the next patch in approximately one-third of all instances. We estimated the average task-relevant travel time as the geometric mean of task-relevant travel times for each animal on each virtual track length. As expected, task-relevant times increased with virtual track length in both low- and high-stochasticity environments, demonstrating that virtual inter-patch distance altered the temporal cost of traveling to the next patch (Fig 6D; average task-relevant travel time: 60 cm, 13.22 s; 100 cm, 16.14 s, 200 cm, 30.14 s). Once they had traversed the virtual track length, animals slowed sufficiently to enter the patch in a time proportionate to the track length (S4E Fig; geometric mean of delay (fraction of average task-relevant travel time): 60 cm, 4.46 s (0.34); 100 cm, 4.77 s (0.30); 200 cm, 7.26 s (0.24)).

In contrast to the freely moving task, in which animals actively nose-poked to remain in a patch, the head-fixed task did not necessitate active engagement while in a virtual patch. Therefore, we considered lick rate to reflect engagement and estimated task-relevant residence time as the time in which lick rate exceeded a minimum threshold (0.5 Hz; S4A–S4B Fig). Mice were engaged in at least 95% of the residence time in 49% of included patches; at least 80% in 61% of included patches; and at least 60% in 66% of included patches. We further analyzed only those patches exceeding 60% engagement for the remainder of the analysis, in order to effectively exclude task-irrelevant behavioral epochs (S4C Fig).

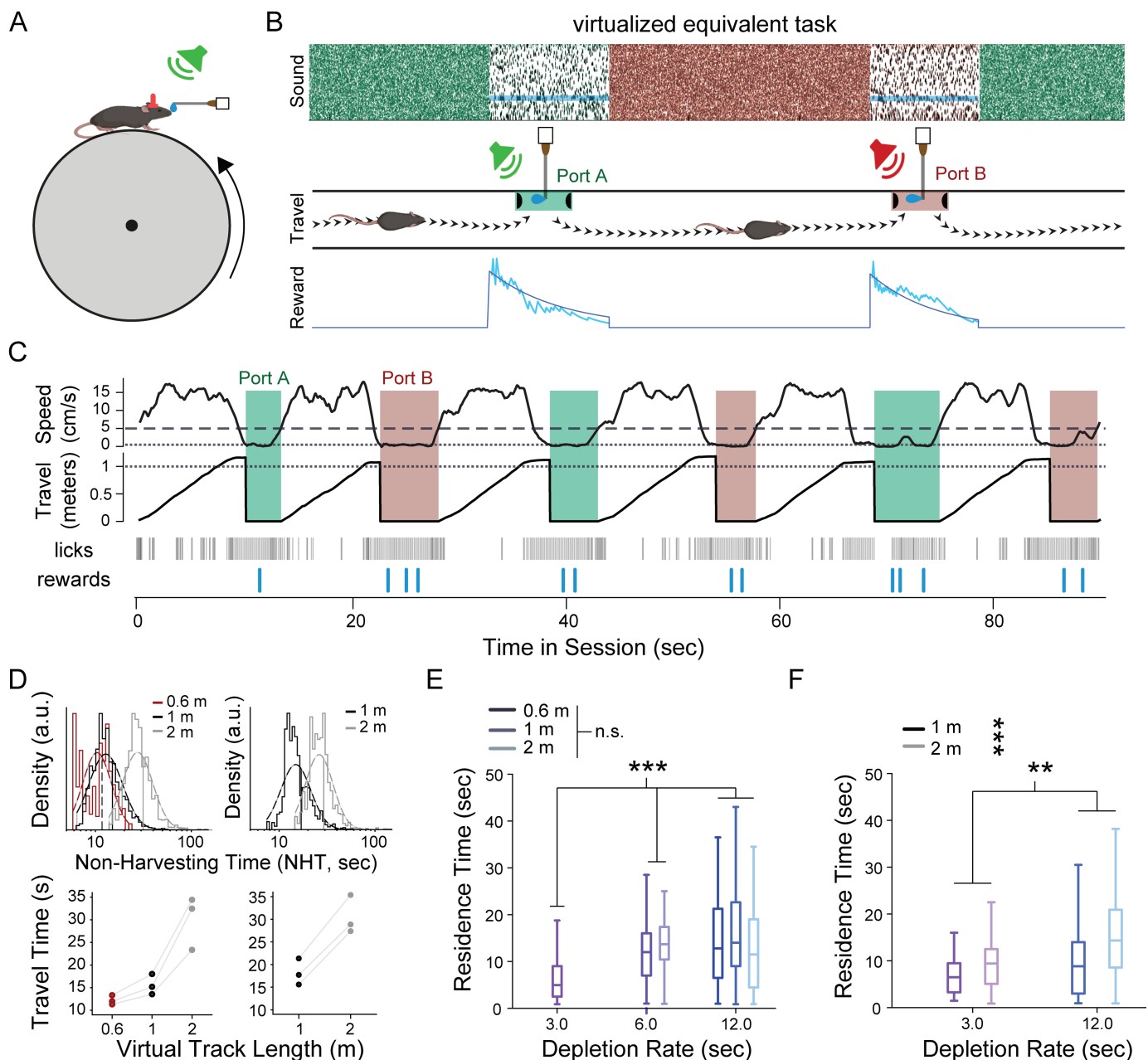

**Fig 6. Virtual patch-based foraging in head-fixed environments followed MVT principles.** (**A**) Head-fixed animals ran on a cylindrical treadmill while receiving auditory cues about the patch location and reward availability. Rewards were dispensed via a lick spout placed in front of the animal. (**B**) Treadmill location was mapped into a 1D virtual space in which patches were separated by a fixed length throughout the session. While stopped in a patch, animals could lick for rewards that became available through a modified Poisson process with an exponentially depleting rate. Tone cloud played continuously, and pure tones played intermittently to indicate the availability of reward(s). As animals ran in between patches, pink noise grew in intensity until the next patch was reached in virtual space, at which time tone cloud began to play again. Once animals had additionally become stationary, they could begin to receive rewards from the new, replenished patch as before. (**C**) Example session with a virtual track length of 1 m and reward decay rate of 3 seconds. Traces for smoothed treadmill speed (top) and corresponding 1D position (middle), as well as raster plots for lick and reward times (bottom), are shown over a 90-second window. Shaded areas correspond to residence times. Dotted lines represent the patch entry criteria for velocity (0.5 cm/s, top) and distance (1 m, middle), while the dashed line represents the velocity threshold for patch exit (5 cm/s, top). Note that the velocity criteria remained the same for all sessions. (**D**) At top, the histogram (solid lines) and corresponding log-normal distribution (dotted lines) of task-relevant travel times is shown for low- (*RSI* = 0.05; left) and moderate-/high- (*RSI* ∈ [0.5, 1.0]; right) stochasticity environments for three virtual track lengths. At bottom, the average task-relevant travel time in low- (left) and moderate-/high- (right) stochasticity environments was computed for each animal on each virtual track length as the geometric mean of the corresponding task-relevant travel times. (**E-F**) A comparison of task-relevant residence times for different environmental parameters is shown for low- (**E**) and and moderate-/high- (**F**) stochasticity environments. Boxes represent the interquartile range (IQR) of residence times from all animals in the given environment. Whiskers extend 1.5x the IQR from the box edges. Center lines represent the median. (n.s.: not significant; ∗∗: *p*<0.001; ∗ ∗ ∗: *p*<0.0001; cluster bootstrap analysis)

We then investigated the effect of environmental parameters on task-relevant residence times in the virtual patch-based foraging task. In low-stochasticity environments ($N$ = 3 mice, 13.0 $\pm$ 2.2 sessions per animal), cluster bootstrap analysis showed that reward decay rate, but not virtual track length, was significantly correlated with task-relevant residence time (Fig 6E; decay rate: $r$ = 0.41 [0.34, 0.48] (mean [95% CI]), $p(r > 0) > 0.9999$; track length: $r$ = 0.01 [−0.08, 0.11], $p(r > 0)$ = 0.59). In high-stochasticity environments ($N$ = 3 mice, 9.7 $\pm$ 2.5 sessions per animal), both parameters were significantly correlated with task-relevant residence time (Fig 6F; decay rate: $r$ = 0.24 [0.12, 0.35] (mean [95% CI]), $p(r > 0)$ = 0.9999; track length: $r$ = 0.36 [0.25, 0.46], $p(r > 0) > 0.9999$). Linear mixed models that predicted task-relevant time from reward decay rate and average task-relevant travel time were in agreement with the cluster bootstrap results (Table B in S1 Table). The signs of the coefficients for all significant parameters were consistent with MVT; that is, slower decay rates or longer travel times correlated with longer residence times. Consequently, average behavior overall reflected the principles of MVT in a head-fixed, virtual patch-based foraging environment. These results demonstrate that patch-based foraging can be implemented in head-fixed paradigms.

## Discussion

Foraging is a canonical choice process in nature and thus, by necessity, has driven the evolution of cognitive processes. As such, the mental pathways utilized during foraging directly correlate to decision-making as it exists in the natural world [4,43,44]. Extensive research in behavioral ecology and ethology have shown that sufficient gathering of resources during foraging tasks may be captured by a variety of strategies [6,15,16,36–38,45,46] or behavioral models [25,38,47–52]. Moreover, because several components of foraging, such as the trade-off between exploration and exploitation, are disrupted in psychiatric disorders [13,14], foraging tasks present a potential cross-species tool for characterizing underlying neural pathology. In contrast, traditional laboratory tasks, which are designed for ease of analysis and interpretation, indirectly shed light on real-world decision-making processes. Importantly, the layers of uncertainty inherent to natural processes are largely absent from traditional task designs and behavioral models. Here, we leveraged a large collection of behavioral data from a novel experimental paradigm to elucidate mechanisms by which animals behave in naturalistic settings. Our results demonstrated that animals modeled meta-uncertainty by combining information on multiple time scales. Rather than solely relying on either average statistics or recent observations, animals utilized a hierarchical framework to harvest resources effectively, simultaneously parsing uncertainty both within and across different distributions of in-patch dynamics.

The uncertainty we modeled in this study occurred at two distinct levels: the uncertainty of the depletion rate of the patch, which resulted from the daily perturbation of the environmental parameter; and the uncertainty of the reward times given a known depletion rate, which resulted from the underlying stochasticity of the reward-generating process. This variability in variability is known as "meta-variability," and the associated cognitive load is known as "meta-uncertainty". To contend with this phenomenon in natural settings, animals must determine, when faced with a deviation from their expectation, whether it is due to variability of the same underlying distribution or instead represents a change in the environmental parameters. Although the former has been studied extensively [18,53], little is known about inherent mechanisms to handle meta-uncertainty. Our experimental design uniquely both introduced and modeled foraging decisions in these stochastic environments, illustrating that animals can build complex models to make informed decisions in challenging, volatile environments. Additionally, rather than needing to compute exact Bayesian estimations of the

underlying parameters, animals instead continually updated their models with recent observations, an approach that provides a mechanistic understanding of foraging in unstable naturalistic settings [54]. Thus our results not only demonstrate that animals have the ability to make decisions effectively in environments with multiple layers of uncertainty, but they can also build remarkably efficient models to perform in such settings, a finding important for perspectives in both behavioral ecology and neuroscience.

Meta-variability is ubiquitous in the natural world and, consequently, has been framed in a number of ways across various disciplines. For example, in the realm of machine learning [55], meta-uncertainty has been theorized to consist of aleatoric uncertainty, which relates to the observed outcome, and epistemic uncertainty, which relates to the model parameters. In the context of the Bayesian model presented here, these two types of uncertainty directly relate to the likelihood (reward timing) and posterior distribution (reward decay rate), respectively [56]. Within the neuroscientific community, meta-uncertainty has been studied in contexts ranging from fluctuations in neural states that encapsulate levels of stimulus variability [57,58] to the uncertainty about confidence in a decision [59]. More generally, a related paradigm introduced by [60] distinguishes between the expected uncertainty associated with inherently stochastic observations (or rewards) and the unexpected uncertainty that arises from a change in the parameters of the underlying process, a concept often tied to reinforcement learning and exploration/exploitation tradeoffs [61–66]. Studies have shown that mice learn to handle expected uncertainty by estimating the variance of a stimulus [67] and, to a lesser degree, they handle unexpected uncertainty by encoding deviations from the expected distribution [68–71]. However, these studies often use traditional, trial-based methods, even in the context of foraging [70]. Furthermore, these methods, such as probabilistic reversal learning, model the response to unexpected uncertainty as animal preference amongst two or more reward sites without explicit inference of the underlying parameters. By contrast, the meta-uncertainty introduced by our task assesses decision-making in both continuous time (variability of reward times) and continuous task space (variability of reward decay rates). Our work builds upon previous models of uncertainty in mice by demonstrating their ability to handle meta-uncertainty in a naturalistic form—patch-based foraging. Consequently, our results facilitate the study of a cognitive repertoire, and underlying neural circuits, which cannot be directly assessed in trial-based tasks.

More generally, the behavioral strategies, and underlying neural circuits, for processing meta-variability and meta-uncertainty are largely unknown and actively being studied. Animals look to prior information for guidance [72]; tend to persist with current choices even in the face of contradictory evidence [73–75] (also known as perseverance, or, related to the case of foraging, over-harvesting); and, intriguingly, make decisions that are, at least in part, inherently stochastic [76]. For instance, although average behavior may correlate with a trained, or even ideal, Bayesian observer, individual decisions maintain a significant level of variability that cannot be captured by behavioral models [76], as was seen in our results. High behavioral variability in the context of foraging has been observed in previous studies [3], and related overharvesting may be explained as either a consequence of Bayesian inference [77], particularly for fast-decaying patches, or as a compensatory mechanism to one's behavioral variability [42]. In particular cases, the forager may build multimodal models of stochastic environments and modify residence times accordingly, creating another source of behavioral variability [78–80], though this strategy is unlikely in our task given the randomized reward sequences. At the same time, the utilization of posterior probability is an important feature of foraging decisions; although early theoretical models of stochastic environments suggested a heuristic strategy [15], our results are consistent with more recent models that propose a

Bayesian approach [49]. This strategy is important because it shapes how one might investigate the underlying neural circuits. For instance, previous studies have demonstrated the importance of the dorsal anterior cingulate cortex (dACC) in patch-leaving decisions via an integrate-to-threshold mechanism [3]. While activity in the dACC transiently increased during stay-or-leave decisions, the encoding during continual decision-making, and faced with meta-uncertainty, is not known. Additionally, neuromodulators such as dopamine, serotonin, or acetylcholine may track different levels and types of variability [51,81–83] or modulate leaving times [25]. The behavioral and analytical framework for stochastic foraging environments we propose here will allow a better mapping between neural activity and natural behavior.

Our approaches and results point to a number of promising areas for further investigation. Although the number of patch encounters included in the analyses of the freely moving task was large (14,060 patches over 300 sessions), the number of animals ($N = 8$) was insufficient to study between-animal differences in behavior. For instance, individual mice exhibited different sensitivities to reward variability, which may reflect either a continuum of learning rates or a cluster of different strategies altogether. Additionally, future experiments could test what aspects of behavior, and underlying neural circuit mechanisms, are common or distinct between freely moving and virtual patch foraging environments. In both tasks, animals also displayed a significant proportion of task-irrelevant behavior, such as exploration or grooming outside of patches in the freely moving task, or periods of inactivity in the head-fixed task, which may have partially resulted the behavioral freedom granted by the naturalistic task design. In the freely moving task in particular, explorations manifested as rearing, sniffing, or attempting to climb the walls of the experimental apparatus; more deliberate behavior, such as changes of mind or head-swivels, was displayed on shorter time scales while nose-poked. Although this layer of behavioral noise was excluded from analysis in this study, future work could utilize more complex data, such as video tracking of animal movements and pupil size, respectively, to build a more comprehensive model of behavior, including both foraging and non-foraging states [41]. Finally, because animals underwent a prolonged training period, and environmental parameters were modified sequentially across days, task learning was difficult to study. Moreover, comparable sessions for the same animal to study a parameter of interest (e.g. $\tau = 12.0$ vs. $\tau = 3.0$ on a given track) were in some cases separated by multiple days due to the experimental schema. Future work using within-session switching of environment parameters, deterministically or stochastically, would allow further elucidation of the time course of behavioral adaptation to environmental perturbations.

Many studies of cognition take a top-down approach in which experiments are designed to test a particular cognitive function. Not only does this lead to design of unnatural tasks, but it also presumes that mental processes derived from human psychology are applicable to the animal species of study. In both cases, the results may be difficult to interpret or even misleading. We instead followed a bottom-up approach by constructing a task from natural principles, allowing animals to perform the task freely, and following their behavior to generate interpretive models. Of course, no laboratory task can perfectly replicate a native environment, and trade-offs must be made between data acquisition (including neural data) and freedom of behavior. There is, however, a growing consensus that laboratory tasks will yield the best results of mimicking the world for which the brain evolved, an idea that is especially true for patch-based foraging [4,5]. Moreover, the behavioral paradigms and analysis approaches outlined in this study provide a framework for investigating further aspects of decision-making processes, such as contextual information or continuous models [84,85]. By invoking native behaviors and presenting naturalistic uncertainty, patch-based foraging tasks offer myriad opportunities to study fundamental decision-making processes.

## Methods and materials

### Ethics statement

All animal procedures were in strict compliance with the ethical guidelines of the National Institutes of Health and approved by the Institutional Animal Care and Use Committee at Baylor College of Medicine. Animal health and well-being were closely monitored for the entirety of the study, and comprehensive efforts were made to minimize suffering.

### Animals

Sixteen male C57/BL6 mice (Jackson Labs) were selected for experiments. Mice began training at 8 weeks of age and performed experimental tasks at 12–24 weeks of age (freely-moving cohort) or 12–20 weeks of age (head-fixed cohort). Mice were restricted to no less than 80% of normal body weight and were given free access to water in their home cages. Mice were kept on a regular light/dark cycle and performed all tasks during the light phase.

### Head-post implantation

Eight of the sixteen mice underwent the procedure for head-post implantation in order to train on the head-fixed experimental setup. All surgical instruments were sterilized prior to use. Animals were anesthetized with isoflurane gas (2–3% in oxygen) for the duration of the procedure. The surgical site was shaved and prepped with betadine and alcohol. An incision was along the midline of the scalp. After the overlying fascia was removed, the skull was scraped and cleaned with a sterile saline solution. A sterile head-post was secured with dental acrylic slightly ( 0.5 cm) anterior to bregma.

### Experimental setup

**Freely-moving behavioral apparatus.** Freely-moving experiments were conducted inside an enclosed sound booth (Otometrics; Schaumburg, IL) in a dark environment. Mice performed the behavioral task on one of two elevated tracks ( 6 cm wide), consisting of a single 100 *cm* segment or two two-meter segments joined at a 90 degree angle (400 *cm* track). Both tracks were lined with red semi-transparent acrylic walls (3 mm x 30 cm; TAP Plastics) to discourage irrelevant exploratory behaviors. Custom 3D-printed reward ports were placed at the ends of the tracks and housed a lick spout (blunt-tip 19G 1.5" needle) centrally. Rewards were dispensed via a syringe pump (model NE-500; New Era Pump Systems, Inc.) that was elevated to the same height as the track to avoid unintentional leakage. Speakers (ES-1 Free Field Electrostatic Speaker; Tucker-Davis Technologies) were mounted approximately 7 cm above each reward port. Speaker output was recorded to the host computer via a custom microphone adaptor board. Overhead webcams (C-920; Logitech) modified to remove the factory infrared (IR) filter recorded experimental activity, which was illuminated by IR illuminators (850nm; Univivi).

Data acquisition and behavioral logic were managed by a custom Python-based behavioral platform (available here). Briefly, the system peripherals consisted of 1) a custom IR beam break circuit to detect pokes within the reward port; 2) a custom capacitive sensor board to detect licks that employed open-source firmware (available here) [86,87]; and 3) an interface with the syringe pump to trigger reward disbursement. The poke and lick detector inputs, and the syringe pump outputs, were managed by a custom I/O board that recorded logic states at 500 Hz and interfaced with the host computer. The digital inputs, audio waveforms, and

video were synchronized via custom Python software running on the host computer. Additional Python software managed the task logic to coordinate the audio and reward outputs as described in the task below.

**Head-fixed behavioral apparatus.** Head-fixed experiments were conducted inside individual sound booths (Otometrics; Schaumburg, IL). Mice ran on a cylindrical treadmill while fixed to the head-post. Rewards were dispensed through a lick spout (blunt-tip 19G 1.5" needle) placed slightly anterior to the mice via a programmable syringe pump (model NE-500; New Era Pump Systems, Inc.). Licks were detected via an electrical sensor (Janelia) connected to the spout. Auditory stimuli played through a speaker (ES-1 Free Field Electrostatic Speaker; Tucker-Davis Technologies) mounted on the left side of the animal. Speakers were calibrated routinely throughout the duration of the study.

Behavioral logic and data acquisition were managed through custom LabVIEW software. Analog outputs from the sound waveform, lick detector, and syringe pump were simultaneously recorded to a DAQ (National Instruments). Treadmill position and velocity was recorded from an encoder (Model 15T Accu-Coder) attached to the treadmill.

**Auditory stimuli.** Auditory stimuli were generated using custom Python (freely-moving) or LabVIEW (head-fixed) code at 192 kHz for playback. Pure tones indicated reward availability. For every available observable reward, the frequency was increased by two semitones, with the base frequency $f_0$ indicating the presence of a single reward. Thus the tone frequency to indicate $n$ available rewards is given by:

$$f_n = f_0 2^{\frac{2n}{12}}$$

Tone cloud stimuli consisted of repeating chords divided into 20 ms bins [88]. Each chord was comprised of 5 semitones randomly selected between 1.5 *kHz* and 96 *kHz*. To reduce boundary anomalies, cosine gating was applied to the first and last 5 ms of each time bin. Pink noise was generated via the Voss-McCartney algorithm using 16 sources [89–91]. Additionally, in the head-fixed task, the intensity of the pink noise, which was played when the animal was in between patches, was modulated according to the inverse square law to mimic natural acoustic attenuation in physical environments. If an animal is some distance $r_1$ from a sound source (e.g. speaker), then the sound intensity $I_1$, sound pressure amplitude $p_1$, and sound pressure level $L_1$ (in decibels) can be approximated as:

$$I_1 = \frac{P}{4\pi r_1^2} \sim p_1^2$$

$$L_1 = 10 log_{10} \frac{p_1^2}{p_0^2} = 20 log_{10} \frac{p_1}{p_0}$$

where $P$ is power and $p_0$ is the reference pressure amplitude. For a given pressure level $L_1$, the pressure level $L_2$ at distance $r_2$ from the sound source is:

$$L_2 = L_1 - 20 log_{10} \left( \frac{p_1}{p_2} \right)$$

$$= L_1 - 20 log_{10} \left( \frac{r_2}{r_1} \right)$$

In the head-fixed task, $r_1$ and $L_1$, the distance to and pressure level of the virtual sound source when the animal was in a patch, was set to 5 $cm$ (the approximate distance in the freely-moving task) and 50 $dB$, respectively. As the animal was approaching a patch, the remaining travel distance, $r_2 - r_1$, was used to calculate the level of attenuation per the equation above. The sound pressure level of the tone cloud stimulus, which played when the animal was stopped in a patch, remained constant at $L_1$.

**Reward dynamics.** Each patch featured rewards that depleted as the animal remained in it. Rewards were always given as 2 $\mu L$ droplets. Because reward volume was fixed, depletion was realized by increasing the interval between rewards over time in patch. The rate at which the inter-reward interval increased, and thus reward rate decreased, was governed by the decay rate parameter ($\tau$), which corresponds to the time constant of the exponential depletion. A larger $\tau$ means intervals increase more slowly, and thus more reward can be harvested in a given interval. In order to ensure that rewards were not delivered with deterministic inter-event intervals, we used a modified Poisson process, known as an inhomogenous gamma process [92], which is described next.

Within a patch, the times at which fixed-volume reward droplets were given followed an inhomogeneous gamma process with an exponentially decaying event rate. Here, we use the term *event* to mean an occurrence in the underlying process, and the term *reward* to mean the observed, 2 $\mu L$ droplet that the animal receives. Because variance in a traditional Poisson process is equal to the expected value over a given interval, the stochasticity and, in this case, reward rate are inextricably linked. For instance, slowing reward depletion (by increasing $\tau$) would increase both the expected number of rewards and the variance of rewards in a patch. However, this would confound analyses of both the reward decay rate and stochasticity, since any change in behavior in response to one could not be separated from a change in response to the other. In order to separate changes in stochasticity from changes in reward decay rate, we instead generate events from a hidden, inhomogeneous Poisson process with an exponentially decaying Poisson rate. Each event is assigned some volume $V_0$, which remains fixed for a given session, and rewards are given whenever the sum of event-volumes exceed the reward droplet volume, $V_r = 2 \mu L$.

The underlying inhomogeneous Poisson process is characterized by the following time-varying rate and its cumulative probability distribution::

$$\lambda(t) = \lambda_0 e^{-\frac{t}{\tau}}$$
$$\Lambda(t,s) = \int_t^{t+s} \lambda(t')dt' = \lambda_0 \tau \left(1 - e^{-\frac{s}{\tau}}\right)$$

Given a decay rate $\tau$, stochasticity is independently varied by modulating the volume associated with each Poisson event, termed $V_0$. To see why, note that the cumulative reward function in each patch becomes:

$$V(s) = V_0 \Lambda(0,s)$$
$$= V_0 \lambda_0 \tau \left(1 - e^{-\frac{s}{\tau}}\right)$$

with the following expectation and variance:

$$\mathbb{E}[V(s)] = V_0 \Lambda(s)$$
$$var[V(s)] = V_0^2 \Lambda(s)$$

By setting $V_0 \lambda_0 = r_0$ for all values of $V_0$, where $r_0 = 2.5 \, \mu L$ is the same for all experiments, we can scale the initial Poisson rate $\lambda_0$ and event volume $V_0$ such that, for a given $\tau$, all patches maintain the same expected reward value but with variance increasing as $V_0$:

$$\mathbb{E}[V(s)] = (V_0 \lambda_0) \, \tau \left(1 - e^{-\frac{s}{\tau}}\right)$$
$$= r_0 \tau \left(1 - e^{-\frac{s}{\tau}}\right)$$
$$var[V(s)] = V_0^2 \Lambda(s)$$
$$= V_0 (V_0 \lambda_0) \, \tau \left(1 - e^{-\frac{s}{\tau}}\right)$$
$$= V_0 \left(r_0 \tau \left(1 - e^{-\frac{s}{\tau}}\right)\right)$$
$$= V_0 \mathbb{E}[V(s)]$$

Thus $V_0$ directly influences the level of reward stochasticity independently of the decay rate. Rewards were made available to the animal whenever the cumulative volume associated with the hidden Poisson process equaled $V_r = 2 \, \mu L$. In other words, every $L = \frac{V_r}{V_0}$ events constituted an observable reward. This modified process, in which every $L^{th}$ event from a inhomogeneous Poisson process is observable, is known as an inhomogeneous gamma process [92].

Moreover, we defined the *reward stochasticity index*, a measure of environmental uncertainty, as the ratio of the event volume to the observed reward volume:

$$RSI = \frac{V_0}{V_r}$$

Thus $RSI$ was necessarily bounded within the interval (0,1]. Increased $RSI$ reflected increased environmental uncertainty. The set of environmental $RSI$ values was [0.05,0.5,1.0].

## Behavioral task

**Freely-moving foraging task.**  Mice selected for the freely-moving task were initially trained to poke and lick from a single reward port while being confined to the last 25 cm of the track. Rewards consisted of 5 $\mu L$ droplets of 10% sucrose solution and were exponentially distributed in time ($\beta = 2 \, s$, $t \in [0.25, 4]$) to encourage persistence. After animals demonstrated significant poking and licking, they were trained to alternate between two reward ports at opposite ends of the track that had the same reward characteristics. Once alternation accuracy (defined as the fraction of poking decisions in which an animal correctly traveled to the opposite reward port) exceeded 60%, animals proceeded to the main foraging task.

In the main foraging task, animals had to poke into one of two reward ports at either end of the track. Both entering and leaving the reward port required a minimum of 500 $ms$ to avoid registering unintentional movements. Once poked, rewards consisting of 2 $\mu L$ droplets became available through the previously described IGP and were dispensed through the lick spout upon licking. Simultaneously, a tone cloud auditory stimulus played through the speaker located above the reward port to denote it as "active." Animals could unpoke at any time to leave the current reward port, at which point the associated speaker stopped playing the tone cloud stimulus to denote it as "inactive", and move towards the other reward port, where the other speaker began playing a pink noise stimulus. Pokes into the same reward port were ignored and did not yield further rewards. Once poked in the other reward port, the adjacent speaker switched to a tone cloud stimulus, and the animal could receive rewards as before from the IGP reset to the initial values. The alternation pattern continued for the

remainder of the session, which typically lasted 30 minutes. Each poke-unpoke sequence is termed a "patch," while the subsequent movement to the next reward port is termed "travel."

**Head-fixed foraging task.** The head-fixed task mirrored the freely-moving version in a virtual environment denoted by auditory cues. Animals were first placed on the cylindrical treadmill and secured to the head-post. The task began with the animals in a virtual "patch", during which a tone cloud auditory stimulus was played. Similar to the freely-moving task, fixed-volume rewards consisting of 2 $\mu L$ droplets (10% sucrose solution) were generated by the underlying modified Poisson process. A pure tone played when reward(s) were available to harvest. Animals could receive the available reward(s) by licking the spout. Patch-leaving decisions were determined by the onset of running, which was defined as treadmill velocity greater than 5 $\frac{cm}{s}$. Once velocity exceeded the running threshold, pink noise played to indicate that the animal was in between patches and rewards were no longer available. In order to enter the next patch, the animal had to traverse a set virtual track length on the treadmill. As the animal approached the next patch, the intensity of the pink noise stimulus grew proportionate to inverse square of the remaining distance, mimicking the inverse square law for acoustics. Once the animal had covered the virtual track distance, tone cloud again played to indicate the animal was in a virtual patch. However, the reward-generating process did not start until the animal had additionally stopped moving, which was defined as velocity less than 0.5 *cm/s*. Note that two different velocity thresholds were used both to 1) avoid rapidly fluctuating in and out of patches and 2) encourage animals to lick while stationary. Treadmill velocity was computed as a running average of over the previous one second and continually monitored for the appropriate threshold crossing.

Animals were first acclimated to the head-fixed apparatus for several days. They then trained on the task with a slow reward decay rate ($\tau$ = 30.0) and short virtual track (15 *cm*) for one week, followed by a faster decay rate ($\tau$ = 12.0) for an additional week. They then performed the foraging task with the environmental parameters described below.

**Task environments.** Each task environment was defined by two reward dynamic parameters ($V_0$, $\tau$) and the track type (physical or virtual) (Table C in S1 Table). The decay rate $\tau$ was varied weekly for both tasks, and the track length was varied daily and weekly for the freely-moving and head-fixed tasks, respectively. Experiments were first conducted with *RSI* = 0.05 until all environments (i.e. $\tau$-track pairs) had been tested, followed by *RSI* = 0.5 and *RSI* = 1.0. Note that fewer values of the decay rate were explored with the larger values of *RSI* due to the large number of potential combinations.

## Data analysis

**Analysis environment.** All analyses were done using Python 3.7 running on Ubuntu 16.04. The linear mixed models were fit using the *statsmodels* package (v0.12.2).

**Inclusion and exclusion criteria.** After training, eight mice completed a total of 440 sessions on the freely-moving foraging task. Sessions comprised at least 20 patches in order to be included in the analysis. A log-normal distribution was fit to all residence times ($\mu$ = 9.62, $\sigma$ = 5.25), and outliers, defined as more than three standard deviations above or below the mean, were excluded from analysis. Latencies between the generated and experienced reward time occasionally arose due to licking behavior and technical errors. Because reward timing is vital to assessing and responding to the environmental dynamics, patches with one or more latencies exceeding 500 *ms* were excluded, and any session that comprised greater than 10% such patches was excluded entirely. Lastly, sessions that included fewer than 10 patches after application of the above criteria were excluded. The remaining dataset comprised 385 sessions

with 17,877 patches. All subsequent analyses were conducted on the less ($RSI$ = 0.05) or more ($RSI \in [0.5, 1.0]$) stochastic experiments separately unless otherwise specified.

For the head-fixed foraging task, experiments comprised 383 sessions across eight mice after the training period. The same criteria as the freely-moving task, but with different thresholds, were initially applied to the dataset (minimum patches in session: 12; log-normal distribution of residence times: $\mu_{log_{10}}$ = 1.32, $\sigma_{log_{10}}$ = 0.58), except for the reward latency criterion. Additionally, animals with more than 50% of sessions that did not meet the above criteria were excluded entirely from the analysis (five of eight). Of sessions in the remaining three animals, one was excluded due to overactive running, and two were excluded because no rewards were given. Lastly, after estimating the task-relevant residence times from licking behavior (see below), patches with task-relevant residence times that were more than two standard deviations below the mean (log-normal distribution) or with active licking comprising less than 60% of the total residence time were excluded from the analysis. (Two standard deviations below the mean was chosen as the threshold to avoid unreasonably small residence times and provide a more conservative estimate of behavioral changes in the head-fixed task.) The remaining dataset consisted of 2,086 patches from 112 sessions amongst three animals. Analyses were likewise conducted on the less or more stochastic environments independently.

**Residence and travel times.** Residence times during the freely-moving task were defined to start and end after the animal had poked and unpoked, respectively, for 500 *ms* continuously at the reward port. The total travel time was consequently the time between the end of one residence time to the start of the next residence time. However, given that animals also exhibited unrelated behavior while traveling, the task-relevant travel time was estimated for each animal in a particular environment (i.e. decay rate and track type) as the tenth percentile of the distribution of total travel times, which approximately represented the inflection point of the cumulative distribution, or, equivalently, the peak of the density function (S7 Fig).

Residence times during the head-fixed task were defined to start when the animal had both traversed the virtual track length and became stationary, and to end when the animal began to run (see Head-fixed foraging task), which coincided with the reward-generating process. Unlike the freely-moving task, in which the nose poke required animals to actively engage in the task in order to be in a patch, animals displayed periods of inactivity during the head-fixed task both within and outside of patches. Task-relevant residence time was estimated using lick rate as a surrogate for task engagement. Lick rate was computed by counting the number of licks within 500 ms time bins and smoothing the ensuing rate with a Gaussian kernel ($\sigma$ = 2 s). Task-relevant residence times were then calculated by excluding time bins in which the smoothed rate fell below 0.5 Hz. A similar procedure was conducted to estimate task-relevant travel times from treadmill velocity, excluding intervals in which velocity fell below the patch entry threshold (0.5 cm/s).

**Cluster bootstrap.** Statistical tests were utilized to assess the effects of environmental parameters on patch residence times. Traditional statistical tests, however, were inappropriate because 1) variance was significantly different between environments, and 2) residence times were not measured independently due to the repeated nature of the experimental design. (Although repeated-measures ANOVA could account for the latter violation, it cannot handle missing data and loses a significant amount of information by collapsing several hundred data points into a single mean.) Therefore, a cluster bootstrap approach [93], which builds upon the original bootstrap methodology [94,95], was taken to account for the hierarchical nature of the data. The data was organized into the following hierarchical levels:

$$environment \Rightarrow animal \Rightarrow session \Rightarrow patch$$

where *environment* consists of a tuple defined by the three environmental parameters, $(\tau, track, V_0)$. The hierarchical representation can be visualized as a tree data structure, with each node representing unique values (e.g. animal IDs) for a given level (e.g. animals) under the parent node (e.g. environment). The data was first separated into the groups at the *environment* level. Within each group, $N_i$ values at the subsequent levels were sampled with replacement, where $N_i$ is the minimum number of nodes at the $i^{th}$ level within the group, until $N_k$ patches were drawn from each sampled session, constituting a sample of size $(N_1)\cdots(N_k)$. Utilizing the minimum number of nodes across levels ensured that the resultant sample was balanced across potential sources of bias (e.g. animal ID). The process was repeated $M = 10,000$ times for each group to build a bootstrapped sampling distribution upon which statistical tests were conducted.

Separate analyses were conducted for the less ($RSI = 0.05$) and more ($RSI \in [0.5, 1.0]$) stochastic conditions. Sampling distributions were sorted by the parameter of interest ($\tau$ or *track*). The Pearson correlation coefficient was computed for each sample to generate $M$ values, and the resulting mean and 95% confidence intervals were calculated. The presence of the entire confidence interval either less than or greater than zero indicates a significant negative or positive correlation, respectively, of the parameter with residence time (assuming a two-tailed Type I error tolerance of 0.05). For comparisons with two values (e.g. track type), the fraction of sample mean differences greater than zero, an equivalent metric, was also calculated. Similarly, fractions less than 0.025 or greater than 0.975 indicate a significant negative or positive relationship, respectively.

**Linear mixed model.**   Residence times were fit to a linear mixed model of the form:

$$\mathbf{y} = \mathbf{X}\beta + \mathbf{Z}\mu + \epsilon$$

where $\mathbf{y}$ is the observed residence times; $\mathbf{X}$ and $\beta$ are the values and parameters, respectively, of the fixed effects; $\mathbf{Z}$ and $\mu$ are the values and parameters, respectively, of random effects; and $\epsilon$ is noise. Fixed effects included environmental parameters ($\tau$ and *track*) and time-on-task effects. Different metrics of the travel time (task-relevant and total travel time) and time on task (patch number, patch start time) were explored until the model with the lowest Bayesian information criterion score was obtained. Mice constituted the random effects in all models. All model inputs were normalized to lie within $[0, 1]$. Likelihood ratio tests between the full model and reduced model, in which the parameter of interest was excluded, were conducted to determine parameter significance. $\chi^2$ and $p$ values were obtained by comparing the log-likelihood ratio to the $\chi_1^2$ distribution.

**Global behavioral models.**   All analyses of the behavioral models and parameter estimation were conducted for the freely-moving task only.

*Expected reward times.* All behavioral models were constructed to predict patch residence times for each animal, given their particular inputs. Of note, patch-leaving criteria for sequence-based models were often not fulfilled at the observed leaving time, creating a need for predicted, unobserved reward times. Because the specific sequence generated for a given patch could introduce bias, the expected future reward times were instead computed and used as model inputs.

To compute the expectation for reward times $\{T_1, \dots, T_M\}$ after time $t$, note the cumulative distribution function for the time of the $M^{th}$ event, using the transformation $S_M = T_M - t$ and $s = t' - t$ for ease of calculation:

$$F_{S_M}(s; t) = P(S_M \leq s; t)$$
$$= 1 - P(S_m > s; t)$$

$$= 1 - \sum_{m=0}^{M-1} P(M(s) = m; t)$$

$$= 1 - \sum_{m=0}^{M-1} \left( e^{-\Lambda(t,s)} \left( \frac{\Lambda(t,s)^m}{m!} \right) \right)$$

where

$$\Lambda(t, s) = \int_t^{t+s} \lambda(s') ds'$$

Because observing the $M^{th}$ event becomes increasing unlikely as $M$ grows, it is not guaranteed to always be observed: $\lim_{s \to \infty} (F_{S_M}(s; t)) < 1$. Thus, for a given sequence, the cumulative probability can be separated into two components:

$$F_{S_M}(s; t) = \frac{P(S_M < s; t)}{P(S_M < \infty; t)} P(S_M < \infty; t)$$

$$= \tilde{F}_{S_M}(s; t) F_0$$

where

$$F_0 = P(S_M < \infty; t)$$

$$= 1 - \sum_{m=0}^{M-1} e^{-\Lambda(t,\infty)} \frac{\Lambda(t,\infty)^m}{m!}$$

represents the probability that the $M^{th}$ event is observed in a given sequence, and $\tilde{F}$ is the normalized cumulative distribution function. Consequently, the normalized probability density function becomes:

$$\tilde{f}_{S_M}(s; t) = \frac{d\tilde{F}_{S_M}(s; t)}{ds}$$

$$= \frac{1}{F_0} \frac{dF_{S_M}(s; t)}{ds}$$

$$= \sum_{m=0}^{M-1} \frac{1}{m!} \left[ e^{-\Lambda(t,s)} \lambda(t + s) \Lambda(t,s)^{m-1} (\Lambda(t,s) - m) \right]$$

The expectation for $S_M$ when the $M^{th}$ event occurs is found by integrating $s\tilde{f}_{S_M}$ over the domain of $s$:

$$\mathbb{E}_t(S_M) = \int_{-\infty}^{\infty} s\tilde{f}_{S_M}(s; t) ds$$

$$= \int_0^{\infty} s \left( \frac{1}{F_0} \sum_{m=0}^{M-1} \frac{1}{m!} \left[ e^{-\Lambda(t,s)} \lambda(t + s) \Lambda(t,s)^{m-1} (\Lambda(t,s) - m) \right] \right) ds$$

$$= \frac{1}{F_0} \sum_{m=0}^{M-1} \left( \frac{1}{m!} \int_0^{\infty} s \left[ e^{-\Lambda(t,s)} \lambda(t + s) \Lambda(t,s)^{m-1} (\Lambda(t,s) - m) \right] ds \right)$$

Lastly, due to the nature of the inhomogeneous Poisson process, some unobserved events may have occurred between the last observed event and the patch-leaving time. To account

for this phenomenon during estimation of the first future reward time, the expected number of unobserved events at patch-leaving, $L_0$ was estimated. The probability that $m$ unobserved events occurred since the last observed reward was given by:

$$P(M(s) = m \mid M(s) < L; t) = \begin{cases} e^{-\Lambda(t,s)} \dfrac{\Lambda(t,s)^m}{m!} & \text{if } m < L \\ 0 & \text{if } m \geq L \end{cases}$$

where $L = \frac{V_r}{V_0}$ was the threshold at which a reward is given (see Reward dynamics). Summing over all values of $m$ gave the marginal probability $P_0$:

$$P_0 = \sum_{m=0}^{\infty} P(M(s) = m; t)$$

$$= \sum_{m=0}^{L-1} P(M(s) = m; t)$$

$$= \sum_{m=0}^{L-1} \left( e^{-\Lambda(t,s)} \frac{\Lambda(t,s)^m}{m!} \right)$$

which was used to normalize the distribution:

$$\tilde{P}(M(s) = m \mid M(s) < L; t) = \begin{cases} \dfrac{e^{-\Lambda(t,s)} \frac{\Lambda(t,s)^m}{m!}}{P_0} & \text{if } m < L \\ 0 & \text{if } m \geq L \end{cases}$$

$$= \begin{cases} \dfrac{e^{-\Lambda(t,s)} \frac{\Lambda(t,s)^m}{m!}}{\sum_{m=0}^{L-1} \left( e^{-\Lambda(t,s)} \frac{\Lambda(t,s)^m}{m!} \right)} & \text{if } m < L \\ 0 & \text{if } m \geq L \end{cases}$$

The expected value was calculated by:

$$E_t[M(s) \mid M(s) < L] = \sum_{m=0}^{L-1} m\tilde{P}(M(s) = m; t)$$

$$= \frac{\sum_{m=0}^{L-1} \left( m e^{-\Lambda(t,s)} \frac{\Lambda(t,s)^m}{m!} \right)}{\sum_{m=0}^{L-1} \left( e^{-\Lambda(t,s)} \frac{\Lambda(t,s)^m}{m!} \right)}$$

$$= L_0$$

The first future observed reward was thus an estimation of $L - L_0$ events, whereas all subsequent future rewards were estimations of $L$ events.

*Local heuristic models.* The elapsed time model (HEU-ETR) predicted residence time based on an animal's average delay between receiving a reward and leaving the patch. First, the mean duration between the last observed reward and patch-leaving time for each animal was calculated:

$$\Delta t_n = \begin{cases} t_n^{(p)} - t_{n,M_n}^{(r)} & \text{if } M_n > 0 \\ t_n^{(p)} & \text{else} \end{cases}$$

$$\overline{\Delta t} = \frac{1}{N} \sum_n \Delta t_n$$

where $n = 1, \ldots, N$ is the patch number, $t_n^{(p)}$ is the $n^{th}$ residence time, and $t_{n,M_n}^{(r)}$ is the last reward time in the $n^{th}$ patch with rewards $m_n = 1, \ldots, M_n$. The predicted residence times for an animal were calculated by first finding the earliest inter-reward interval that was greater than the leaving criterion $\overline{\Delta t}$:

$$m_n^* = \min \left\{ m_n \mid \tilde{t}_{n,m_n+1}^{(r)} - \tilde{t}_{n,m_n}^{(r)} \geq \overline{\Delta t} \right\}$$

where $\tilde{\mathbf{t}}_n^{(r)} = \left[ (\mathbf{t_n}^{(r)})^T, (\hat{\mathbf{t}}_n^{(r)})^T \right]^T$ is the concatenation of the observed reward times $\mathbf{t_n}^{(r)}$ and the expected future reward times $\hat{\mathbf{t}}_n^{(r)}$. The predicted residence times were then calculated as the time of reward $m_n^*$ followed by the average patch-leaving delay:

$$\hat{y}_n = \tilde{t}_{m_n^*}^{(r)} + \overline{\Delta t}$$

To estimate residence time based on observing a certain number of rewards (HEU-NR), the mean number of rewards observed at patch-leaving was similarly computed for each animal:

$$\overline{M} = \frac{1}{N} \sum_n M_n$$

and using the same framework for constructing observed and future reward times, residence times were predicted as the time at which reward $\overline{M}$ was observed in the patch:

$$\hat{y}_n = \tilde{t}_{\overline{M}}^{(r)}$$

*Marginal value theorem.* According to the marginal value theorem (MVT), the animals should leave the patch when the instantaneous rate of return, $v(t)$, equals or falls below the average rate of return in the environment:

$$v(t^{(p)}) = \frac{V(t^{(p)})}{t^{(p)} + t^{(t)}}$$

Given the equations for $v(t)$ and $V(t)$ above, the value $t^{(p)\star}$ that satisfies this condition, which is the optimal residence time according to MVT (MVT-OPT), is:

$$e^{-\frac{t^{(p)\star}}{\tau}} \left( t^{(t)} + t^{(p)\star} + \tau \right) - \tau = 0$$

for a given set of values $(\tau, t^{(t)})$ defined by the environment. Note that according to MVT, the optimal residence time is independent of the initial reward rate. If the $k^{th}$ environment has associated parameters $(\tau_k, t_k^{(t)})$, then the predicted residence time for a patch $n$ in environment $k$ is:

$$\hat{y}_n = t_k^{(p)\star}$$

where $t_k^{(p)\star}$ satisfies the previous equation for $(\tau_k, t_k^{(t)})$. Predictions for each animal were based on the known value $\tau_k$ and the travel time that was estimated from all sessions for that animal on the track in environment $k$. Optimal residence times were calculated by applying Broyden's first Jacobian approximation to solve for $t_k^{(p)\star}$.

The internal MVT model (MVT-IM) presumes the same underlying presumptions but allows for different perceived parameters to fit the observed data. In particular, the predicted time in environment $k$ is given by $\hat{t}_k^{(p)}$ such that:

$$e^{-\frac{\hat{t}_k^{(p)}}{\hat{\tau}_k}} \left( \hat{t}_k^{(t)} + \hat{t}_k^{(p)} + \hat{\tau}_k \right) - \hat{\tau}_k = 0$$

The parameters $\hat{\tau}_k$ and $\hat{t}_k^{(t)}$ were fit to the observed data by minimizing the following loss function:

$$L(\theta) = \| \hat{\mathbf{y}} - \mathbf{y} \|_2^2 + \lambda \left\| \hat{\theta} - \theta \right\|_2^2$$

where $\mathbf{y} = [t_1^{(p)}, \ldots, t_N^{(p)}]^T$ and $\hat{\mathbf{y}} = [\hat{t}_1^{(p)}, \ldots, \hat{t}_N^{(p)}]^T$ are the observed and predicted residence times, respectively, and $\theta = \{\tau, \mathbf{t}^{(t)}\}$ and $\hat{\theta} = \{\hat{\tau}, \hat{\mathbf{t}}^{(t)}\}$ are the experimental and perceived environmental parameters, respectively. The regularization term $\hat{\theta}$ ensured that the fitted parameters maintained reasonable proximity to the observed values. Additionally, constraints were imposed on the parameters such that the number of fitted and experimental parameters remained equal:

$$\hat{\tau}_k = \hat{\tau}_{k'} \quad \forall \quad k, k' \quad \text{s.t.} \quad \tau_k = \tau_{k'}$$
$$\hat{t}_k^{(t)} = \hat{t}_{k'}^{(t)} \quad \forall \quad k, k' \quad \text{s.t.} \quad d_k = d_{k'}$$

where $d_k$ indicates the track length in environment $k$. In other words, in a given dataset (comprised of a given stochasticity level on either the freely-moving or head-fixed task), each value for the experimental $\tau$ corresponded to one estimate $\hat{\tau}$, and each value for the track length corresponded to one estimate for travel time, $\hat{t}^{(t)}$. The parameters were fit using the Broyden–Fletcher–Goldfarb–Shanno algorithm to minimize the loss function.

**Local behavioral models.**

*Parameter estimation.* Bayesian estimates of the reward rate were derived from similar principles shared by previous models [77] but adapted to the specific reward structure of the task. Given a series of patches, $m = 1, \ldots, M$, and reward times within those patches, $[t_1, \ldots, t_{K_m}]$, constituting the inhomogeneous gamma process, the probability of the Poisson rate $\lambda$ at time $t$ is proportional to:

$$p(\lambda | x(t); \lambda_0, \tau) = \prod_m \prod_k \lim_{dt \to 0} \frac{1}{dt} \left( p(N = L - 1 \in \{t_{k-1}, t_k\}) p(N = 1 \in \{t_k, t_k + dt\}) \right)$$

$$= \prod_m \prod_k \left( e^{-\Lambda(t_{k-1}, t_k)} \frac{(\Lambda(t_{k-1}, t_k))^{L-1}}{(L-1)!} \right) (\lambda(t_k) dt)$$

and the log-probability is proportional to:

$$ln(p(\lambda | x(t); \lambda_0, \tau)) = \sum_m \sum_k \left( - \Lambda(t_{k-1}, t_k) + (L-1) ln(\Lambda(t_{k-1}, t_k)) - ln((L-1)!) \right.$$

$$\left. + ln(\lambda(t_k) + ln(dt)) \right)$$

$$= \sum_m \left( - \Lambda(t_m) + (L-1) \sum_k ln(\Lambda(t_{k-1}, t_k)) - K_m ln((L-1)!) \right.$$
$$\left. + \sum_k \left( ln(\lambda(t_k)) + ln(dt) \right) \right)$$

where $L = \frac{V_r}{V_0} = \frac{1}{RSI}$ is the number of Poisson events that constitute an observable reward, and $\lambda(t)$ and $\Lambda(t)$ are as defined previously. (Here, $\Lambda(t)$ is shorthand for $\Lambda(0, t)$.) The maximum likelihood estimate (MLE) of the parameters $(\tau, \lambda_0)$ is found by setting their partial derivatives equal to zero:

$$\frac{\partial ln(p(\lambda | x(t); \lambda_0, \tau))}{\partial \lambda_0} = 0 = - \sum_m \tau \left( 1 - e^{\frac{-t_m}{\tau}} \right) + L \frac{\sum_m K_m}{\lambda_0}$$

$$\frac{\partial ln(p(\lambda | x(t); \lambda_0, \tau))}{\partial \tau} = 0 = -\lambda_0 \sum_m \left( 1 - e^{\frac{-t_m}{\tau}} \right) + \frac{\lambda_0}{\tau} \sum_m t_m e^{\frac{-t_m}{\tau}} + \frac{(L-1) \sum_m K_m}{\tau}$$
$$+ \frac{L-1}{\tau^2} \sum_m \sum_k \alpha(t_{k-1}, t_k) + \frac{\sum_m \sum_k t_k}{\tau^2}$$

where $\quad \alpha(t_{k-1}, t_k) = \left( \frac{t_{k-1} e^{\frac{-t_{k-1}}{\tau}} - t_k e^{\frac{-t_k}{\tau}}}{e^{\frac{-t_{k-1}}{\tau}} - e^{\frac{-t_k}{\tau}}} \right)$

Solving for $\lambda_0$ in the first equation and plugging it into the second, the following equation for $\tau$ is obtained:

$$0 = \left( \sum_m K_m \tau - \sum_m \sum_k t_k - (L-1) \sum_m \sum_k \alpha(t_{k-1}, t_k) \right) \left( \frac{\sum_m \left( 1 - e^{\frac{-t_m}{\tau}} \right)}{\sum_m t_m e^{\frac{-t_m}{\tau}}} \right) - L \sum_m K_m$$

The solution, $\tau_{MLE}$ was found by applying Brent's method to the above equation. Leveraging the relationship $\lambda(t) = \lambda_0 e^{-\frac{t}{\tau}}$, the MLE of the Poisson rate, $\lambda_{MLE}(t)$, was then calculated by rearranging the equation for $\frac{\partial ln(p(\lambda | x(t); \lambda_0, \tau))}{\partial \lambda_0} = 0$ and substituting $\lambda(t)$ for $\lambda_0$:

$$\lambda_{MLE}(t) = \frac{L \sum_m K_m}{\sum_m \tau_{MLE} \left( 1 - e^{\frac{-t_m}{\tau_{MLE}}} \right)} e^{-\frac{t}{\tau_{MLE}}}$$

Errors in parameter estimation were calculated from only the current ($M = 0$) reward sequence. The corresponding changes in residence time were calculated as the deviation from the average of all $N$ residence times in the session:

$$\Delta t_n^{(p)} = t_n^{(p)} - \frac{1}{N} \sum_{i=1}^{N} t_i^{(p)}$$

To compute the change in residence time (but not the rate estimation error), the time-on-task effect was removed from all residence times. A best-fit line relating residence time to the

patch number in a session was calculated for each animal. The change from baseline based on its slope was then added to the residence times for each animal prior to calculating both the session average and deviation from session average. A control dataset was generated by shuffling the residence times across patches within each session. Given the newly assigned residence times, the rate estimation error at patch-leaving and change in residence time were computed for each patch, where the time-on-task effect was removed prior to computing the latter as before.

A bivariate Gaussian distribution was fit to the set of rate estimation errors and their corresponding changes in residence time for both the observed and shuffled data. Data outside of the ellipse representing the 99th percentile were excluded. Linear regression was performed on the remaining data using rate estimation error and change in residence time as the explanatory and response variable, respectively, using five-fold cross-validation. As in the behavioral model assessment, cross-validation subsets were constructed by dividing residence times within each session into five groups, and combining each group over all sessions to build five subsets.

*Predictive model.* The MLE of the Poisson rate, $\lambda_{MLE}$, was utilized to form a predictive model of foraging decisions. Following the theoretical framework of MVT, the model predicted patch-leaving to occur when the estimated reward rate (i.e. $\lambda_{MLE}$) fell below a threshold for a given environment. For a given environment $k$, the threshold $\lambda_k^*$ was derived from the parameters of MVT-IM, $\hat{\tau}_k$ and $\hat{t}_k^{(t)}$, for each animal as:

$$\lambda_k^* = \lambda(\hat{t}_k^{(p)}) = \lambda_0 e^{-\frac{\hat{t}_k^{(p)}}{\hat{\tau}_k}}$$

where $\hat{t}_k^{(p)}$ is the predicted residence time in environment $k$ according to the MVT-IM model. The estimated Poisson rate was evaluated for each patch at 100 ms intervals. The predicted residence according to the MLE-M model was the first time point in which the estimated reward rate was less than or equal to the patch-leaving threshold:

$$\hat{y}_n = min \left\{ t_i \mid \lambda_{MLE}(t_i) \leq \lambda_k^* \right\}$$

where $t_i$ represents the $i$th time bin, and $M$ refers to the number of patches preceding patch $n$ to include in the MLE. (For initial patches with $n < M$, the first $n$ patches were included). Note that for models with $M = 0$ or no observed rewards in sequences prior to a given patch, the MLE for time bins prior to the first observed reward for such a patch trivially yielded a homogeneous process with zero reward rate (i.e. $\lambda_{MLE} = 0$ and $tau_{MLE} = \infty$), which is incongruent with the MVT-based threshold strategy. To address these initial patch times, a very weak prior was incorporated into the model to avoid nonsensical model behavior, as described below. However, the prior had negligible effect on the estimated reward rate, and consequently the predicted patch-leaving time, once either of the criteria had been satisfied.

**Multiscale behavioral models.**

*Parameter estimation.* To provide models with estimates derived globally, prior probabilities for the parameters ($\lambda_0$ and $\tau$) of the inhomogeneous gamma process (IGP) were included with the likelihood to generate *maximum a posteriori* (MAP) estimates of the current reward rate. The gamma distribution was chosen because it is the conjugate prior for the Poisson distribution, allowing the resulting equations to be more computationally tractable. (Due to the inhomogeneity of the gamma process underlying reward timing (i.e. the non-stationary term

$\Lambda(t)$), the gamma prior distribution does not yield a Poisson posterior distribution and thus is not technically a conjugate prior for the IGP, as seen below.)

The prior distributions of both $\lambda_0$ and $\tau$ were of the general form:

$$p(\lambda_0) = Gamma(\alpha_\lambda, \beta_\lambda) = \lambda_0^{\alpha_\lambda - 1} e^{-\beta_\lambda \lambda_0} \frac{\beta_\lambda^{\alpha_\lambda}}{\Gamma(\alpha_\lambda)}$$

$$p(\tau) = Gamma(\alpha_\tau, \beta_\tau) = \tau^{\alpha_\tau - 1} e^{-\beta_\tau \tau} \frac{\beta_\tau^{\alpha_\tau}}{\Gamma(\alpha_\tau)}$$

where the parameters $(\alpha, \beta)$ are the shape and rate parameter, respectively, for the gamma distribution; $\Gamma$ represents the gamma function; and $\lambda_0$ is abbreviated to $\lambda$ for visual clarity. Each of the IGP parameters thus had an independent prior distribution. By incorporating these prior distributions into the general model presented above, the following posterior distribution was generated:

$$p(\lambda|x(t)) \sim p(x(t)|\lambda) p(\lambda_0) p(\tau) = \prod_m \prod_k \left( e^{-\Lambda(t_{k-1}, t_k)} \frac{(\Lambda(t_{k-1}, t_k))^{L-1}}{(L-1)!} \right) (\lambda(t_k) dt)$$

$$\times \left( \lambda_0^{\alpha_\lambda - 1} e^{-\beta_\lambda \lambda_0} \frac{\beta_\lambda^{\alpha_\lambda}}{\Gamma(\alpha_\lambda)} \right) \left( \tau^{\alpha_\tau - 1} e^{-\beta_\tau \tau} \frac{\beta_\tau^{\alpha_\tau}}{\Gamma(\alpha_\tau)} \right)$$

with the corresponding log-posterior:

$$ln(p(\lambda|x(t))) \sim ln\big(p(x(t)|\lambda)\big)\big(p(\lambda_0)\big)\big(p(\tau)\big) = ln\big(p(x(t)|\lambda)\big)$$

$$+ \left( (\alpha_\lambda - 1) ln(\lambda_0) - \beta_\lambda \lambda_0 + \alpha_\lambda ln(\beta_\lambda) - ln(\Gamma(\alpha_\lambda)) \right)$$

$$+ \left( (\alpha_\tau - 1) ln(\tau) - \beta_\tau \tau + \alpha_\tau ln(\beta_\tau) - ln(\Gamma(\alpha_\tau)) \right)$$

where $ln\big(p(x(t)|\lambda)\big)$ was given in the previous section. Analogous to the MLE, the MAP estimate of the IGP parameters ($\lambda_0$ and $\tau$) was calculated by setting the respective partial derivatives of $ln(p(\lambda|x(t)))$ to zero:

$$\frac{\partial ln(p(\lambda|x(t)))}{\partial \lambda_0} = 0 = -\sum_m \tau \left( 1 - e^{\frac{-t_m}{\tau}} \right) + L \frac{\sum_m K_m}{\lambda_0} + \frac{\alpha_\lambda - 1}{\lambda_0} - \beta_\lambda$$

$$\frac{\partial ln(p(\lambda|x(t)))}{\partial \tau} = 0 = -\lambda_0 \sum_m \left( 1 - e^{\frac{-t_m}{\tau}} \right) + \frac{\lambda_0}{\tau} \sum_m t_m e^{\frac{-t_m}{\tau}} + \frac{(L-1)\sum_m K_m}{\tau}$$

$$+ \frac{L-1}{\tau^2} \sum_m \sum_k \phi(t_{k-1}, t_k) + \frac{\sum_m \sum_k t_k}{\tau^2} + \frac{\alpha_\tau - 1}{\tau} - \beta_\tau$$

where

$$\phi(t_{k-1}, t_k) = \frac{t_{k-1} e^{\frac{-t_{k-1}}{\tau}} - t_k e^{\frac{-t_k}{\tau}}}{e^{\frac{-t_{k-1}}{\tau}} - e^{\frac{-t_k}{\tau}}}$$

and $ln\big(p(x(t)|\tau)\big)$ was derived in the previous section. As before, solving for $\lambda_0$ in the first equation and substituting it into the second equation yielded the following equation for $\tau$:

$$0 = \left( L\sum_m K_m + \alpha_\lambda - 1 - (L-1)\sum_m K_m - \frac{L-1}{\tau}\sum_m\sum_k \phi(t_{k-1}, t_k) \right.$$

$$\left. - \frac{\sum_m\sum_k t_k}{\tau} - \alpha_\tau + 1 + \beta_\tau\tau \right)$$

$$\times \left( \frac{\tau\sum_m\left(1 - e^{\frac{-t_m}{\tau}}\right) + \beta_\lambda}{\sum_m t_m e^{\frac{-t_m}{\tau}} + \beta_\lambda} \right) - \left( L\sum_m K_m + \alpha_\lambda - 1 \right)$$

Similarly, the MAP estimate of $\tau$ was computed by applying Brent's method to the above equation, and the MAP estimate of $\lambda$ was subsequently calculated as:

$$\lambda_{MAP}(t) = \frac{L\sum_m K_m + \alpha_\lambda - 1}{\sum_m \tau_{MAP}\left(1 - e^{\frac{-t_m}{\tau_{MAP}}}\right) + \beta_\lambda} e^{-\frac{t}{\tau_{MAP}}}$$

*Predictive model.* As with the ML estimates, a predictive model was built from the MAP estimates of the reward rate using the MVT construct: for a given environment $k$, the predicted patch-leaving time $y_n$ corresponded to the first time point $t_i$ in the patch in which the estimated reward rate $\lambda_{MAP}$ was less than the leaving threshold $\lambda_k^*$. Simplistically, the Bayesian (MLE- or MAP-based) models predict patch-leaving times in two distinct steps: 1) estimate the underlying reward rate parameters, and thus current reward rate, from previous observations, and 2) leave the patch when the estimated reward rate is less than the model threshold. The first step, however, was significantly more computationally expensive than the second, which guided approaches to numerical optimization below.

First, the centers and shapes of the prior distributions were determined. Because they were governed by parameters $(\alpha_\lambda, \beta_\lambda, \alpha_\tau, \beta_\tau)$ in a continuous, four-dimensional space, numerical approaches to optimization based on minimizing predictive error were computationally intractable; every parameter adjustment during an iteration would require recalculation of all MAP estimates for all time points. Therefore, a grid search was instead conducted over a discrete space limited to prior distributions that were centered on the IGP parameter value corresponding to that of the MVT-IM model but differing in variance (Fig 5A). The mode, as opposed to mean, of the prior distribution was chosen to represent the center because in the absence of information from the likelihood function, the MAP estimate of the reward rate simply becomes the mode of the prior (i.e. the maximum). Consequently, the modes of $p(\lambda_0)$ were equivalent for all animals in a given environment $k$, but those of $p(\tau)$, set to $\hat{\boxtimes}$ in the MVT-IM model, varied by animal and environment $k$:

$$mode(p_k(\lambda_0)) = \frac{\alpha_{\lambda,k} - 1}{\beta_{\lambda,k}} = \lambda_{0,k}$$

$$mode(p_k(\tau)) = \frac{\alpha_{\tau,k} - 1}{\beta_{\tau,k}} = \hat{\tau}_k$$

Given the constraints of the equations above, the variance of each point in parameter space was given by:

$$\sigma^2(p_k(\lambda_0)) = \frac{\alpha_{\lambda,k}}{\beta_{\lambda,k}^2}$$

$$\sigma^2(p_k(\tau)) = \frac{\alpha_{\tau,k}}{\beta_{\tau,k}^2}$$

For each animal, each set of prior parameter values in the grid was used to generate MAP estimates of the reward rates for all patches.

For a given animal-environment pair $k$, the leaving threshold $\lambda_k^*$ was either calculated from the parameters of the MVT-IM model (MAP-IM-L), as in the MLE-x model, or fit to the experimental data to minimize prediction error (MAP-IM-GL). In the latter, the best-fit leaving thresholds were computed using the Nelder-Mead algorithm to find iteratively the simplex of leaving thresholds that minimized prediction error for patch-leaving times. Unlike the MVT-IM model, the algorithm had no natural way of constraining the leaving thresholds to eight values (two track lengths, four decay rates) per animal; consequently, each animal was assigned a best-fit leaving threshold per unique environment. However, fitting the MVT-IM to the high-stochasticity environments ($RSI \in [1.0, 2.0]$) similarly without constraints did not significantly reduce its prediction error nor affect the significance of model comparisons (root-mean-square prediction error (RMSE) [95% CI]: unconstrained MVT-IM, $3.96\,[3.79, 4.13]$, constrained MVT-IM, $3.98\,[3.81, 4.16]$). Due to the large computational cost of fitting reward rate thresholds, a grid search of the prior distributions was conducted over narrowed range of values that was centered around the best-fit results from MAP-IM-L (Fig 5A, right); additionally, the search was limited to models that utilized observations from only the current patch ($N = 1$).

**Model comparisons.** All behavioral models were assessed by measures of their predictive error. The mean absolute error (MAE) was calculated as:

$$MAE = \frac{1}{N}\sum_n |\hat{y}_n - y_n|$$

and the root-mean-square error (RMSE) as:

$$RMSE = \sqrt{\frac{1}{N}\sum_n (\hat{y}_n - y_n)^2}$$

Lastly, the $R^2$ value was calculated as:

$$R^2 = 1 - \frac{\sum_n (\hat{y}_n - y_n)^2}{\sum_n (y_n - \bar{y})^2}$$
$$= 1 - \frac{N \times (RMSE)^2}{\sum_n (y_n - \bar{y})^2}$$

All models underwent five-fold cross-validation. Data subsets were generated by splitting each session into five groups of patches of approximately equal length to ensure that all hierarchical levels of the data were equally represented in each data subset. The null model (HEU-CT) predicted residence times to be the average residence time for each animal across

all sessions ($R^2$ = 0 by definition). Mean error metrics were calculated from the average of all errors in all test sets. Confidence intervals were computed by bootstrapping $M = 10,000$ samples of length $N$ from the set of prediction errors, taking the average of each sample to generate a distribution of sample means, and finding the percentiles corresponding to $\left[\frac{\alpha}{2}, 1 - \frac{\alpha}{2}\right]$, with $\alpha$ = 0.05.

## Supporting information

**S1 Fig. Example of various stochasticity levels with equivalent expected reward.** (*top*) Cumulative reward functions for environments with $\tau$ = 12.0 and a low (left), moderate (center), or high (right) level of stochasticity in the reward dynamics, shown as mean ($V_0\Lambda(t)$, solid curve) $\pm$ standard deviation ($V_0^2\Lambda(t)$, shaded area). (*bottom*) Three example reward sequences for each level of stochasticity.
(PDF)

**S2 Fig. Correlations in local information were not present in shuffled data.** Within sessions, residence times were shuffled across patches prior to calculating to rate estimation error at patch-leaving and the change in residence time relative to the session average. The scatter plot, regression line, and marginal distributions were then calculated from the shuffled data as in Fig 4B–4C for environments with (**A**) low (*RSI* = 0.05) and (**B**) moderate to high (*RSI* $\in$ [0.5, 1.0]) stochasticity.
(PDF)

**S3 Fig. Aspects of the Bayesian behavioral models.** (**A**) The root-mean-square prediction errors (RMSE) of various MLE-x models (solid line) were calculated for all patches in environments with moderate-to-high stochasticity (*RSI* $\in$ [0.5, 1.0]). In order to estimate the Poisson rate, models utilized the observations from the current patch plus zero (MLE-1) up to ten (MLE-11) of the previous patch encounters. The prediction error asymptotically approached that of the MVT-IM model (dotted line). (**B**) Two example prior distributions (solid or dotted curve) for the initial Poisson rate ($\lambda_0$; left) and decay rate ($\tau$; right) are shown for three different levels of variance (*var*($p$)). Prior distributions were generated from a gamma distribution such that the mode was equal to either the experimental ($\lambda_0$) or internally-modeled ($\tau$) value for the environment (solid or dotted triangle). (**C**) The Poisson rate estimates of the MAP-IM-L model are shown for an example reward sequence in a patch (raster at bottom), which consists of unobserved (light purple) and observed (dark purple) events. The rate estimates utilized observations from the example patch and reflect prior distributions with high (orange), moderate (light blue), and low (dark purple) levels of uncertainty, as shown in **B**. The predicted leaving times for the models (colored triangles) occur when the estimated rates fall below a given threshold for the environment (dotted black line) that is derived from the MVT-IM model. (**D**) Given the same example reward sequence as in **C**, the MAP-IM-GL model estimates the Poisson rate (purple solid line) from observations (raster at bottom) and additionally modulates the rate threshold for patch-leaving (black dotted line). Higher (dark red) or lower (light red) thresholds lead to earlier or later leaving times, respectively. The true Poisson rate in **C** and **D** is shown by the black curve.
(PDF)

**S4 Fig. Inclusion criteria for the head-fixed task.** (**A**) The raw (solid line) and smoothed (dotted curve) histogram of the smoothed lick rate (bin size = 0.5 seconds, $\sigma$ = 2 seconds) is shown for data pooled from all animals ($N$ = 3) on the head-fixed task. The rate threshold for active engagement (vertical dotted line) was chosen to represent the "elbow" of the second derivative of the smoothed histogram of lick rates (*inset*). (**B**) The animal licks (raster

at bottom) and smoothed lick rate (solid purple curve) are shown for a 200-second window of an example session. The residence time (shaded areas) was estimated as the time during which the smoothed lick rate exceeded the rate threshold (horizontal dotted line). (**C**-**E**) A histogram (step-wise solid line) and fitted log-normal distribution (shaded area) with its associated median (vertical solid line) are shown for the estimated residence times (**C**; calculated per **B**), travel times (**D**; calculated as time between patches during which velocity exceeded 0.5 cm/s), and delay from traveling the required distance to stopping within the next patch (**E**). (*insets*) Histograms (purple bins, left axis) and cumulative summations (solid line, right axis) of the fraction of raw residence (**C**) and travel (**D**) times during which the animal met the respective engagement criteria. Residence times additionally required at least 60% engagement (vertical dotted line in **C**) for the patches to be included in the analysis.
(PDF)

**S5 Fig. Overview of the cluster bootstrap approach.** Residence times, which are the data points of analysis, exist within a hierarchical structure of contextual characteristics that influence outcomes, including the environment (such as travel distance or reward decay rate), animal, or session in which the patch occurred. When sampling via the bootstrap method, these characteristics (colored outlines of circles) must be appropriately randomized at each level to respect their individual contributions to the overall outcomes.
(PDF)

**S6 Fig. Internal models of environmental parameters.** For environments with low- (**A**; $RSI = 0.1$) and moderate-to-high (**B**; $RSI \in [0.5, 1.0]$) stochasticity, the estimates of the reward decay rate (left) and travel time (right) were calculated per the MVT-IM model, which constrained each animal to have a one-to-one mapping between parameter estimates (vertical axis) and unique environments (horizontal axis). Internal estimates are shown for individual animals (colored squares and dotted lines), which were used in the analysis, and for pooled data (black squares and dotted lines), which are shown for visualization purposes. The experimental values are also shown for individual animals (colored circles and solid lines) and pooled data (black circles and solid lines); note that individual experimental decay rates were equivalent (i.e. independent of animal behavior) and thus are not shown.
(PDF)

**S7 Fig. Sensitivity of LMM results to estimation of task-relevant time.** (**A**) The histogram (left) and cumulative distribution (right) of total travel times (defined as the duration between nose-pokes at successive reward ports) across all freely-moving sessions included in the analysis, categorized by track type (1 meter: purple; 4 meters: light blue). The tenth percentiles of the distributions are represented by the dotted lines in both panels. (**B**, **C**) The coefficient values (left) of the linear mixed model fit to data in the low- (**B**) and high- (**C**) stochasticity environments are shown for the decay constant (tau; black), task-relevant travel time (t_t_est; purple), and patch number (n_p; light blue). Solid lines and shaded areas represent the mean and 95% confidence intervals, respectively. The corresponding p-value for the model fits are shown at right. Note that the p-values for the decay constant and patch number remain highly significant and overlap for the entirety of the percentile domain.
(PDF)

**S1 Table. Tables of parameters for linear mixed-effects models and foraging environments.** (**A**) Linear mixed-effects models of the freely moving task. Parameters were fit as predictors of residence time for patches in low-stochasticity ($RSI = 0.05$, $m = 9547$ patches) or high-stochasticity ($RSI \in [1.0, 2.0]$, $m = 4513$ patches) environments. All fixed effects were normalized to the range $[0, 1]$. Coefficient values are provided as mean [95% CI]. $\chi^2$ and $p$

values were generated from likelihood ratio tests between the full model and reduced model with the respective parameter removed. Key: $\tau$ = decay rate, $\hat{t}^{(t)}$ = task-relevant travel time, $n_p$ = patch number. (**B**) Linear mixed-effects models of the head-fixed task. Notation and analysis follows (**A**). Low-stochasticity ($RSI$ = 0.05, $m$ = 1299 patches) and high-stochasticity ($RSI \in$ [1.0, 2.0], $m$ = 787 patches) environments were analyzed separately. (**C**) List of environmental parameters. Reward stochasticity index (RSI) and decay rate are defined elsewhere. Decay rate is given in seconds and track length in meters.
(PDF)

## Acknowledgments

We thank Wenhao Zhang, Anton Banta, and Hong Jiang for their contributions to the development of the head-fixed experimental setup. We are grateful for the assistance with maintaining mouse colonies and conducting experiments from many colleagues, including Matt Thompson, Kit Jaspe, Hannah Ramsaywak, Jack Shi, Marisa Hudson, and the team of undergraduate students. We also wish to acknowledge Zakir Mridha, who generously assisted with mouse head-post implantation, and Sibo Gao, who co-developed the software for conducting the freely moving experiments.

## Author contributions

**Conceptualization:** James Webb, Daeyeol Lee, Caleb Kemere, Matthew McGinley.

**Data curation:** James Webb.

**Formal analysis:** James Webb, Caleb Kemere, Matthew McGinley.

**Funding acquisition:** Caleb Kemere, Matthew McGinley.

**Methodology:** James Webb, Caleb Kemere, Matthew McGinley.

**Project administration:** Caleb Kemere, Matthew McGinley.

**Software:** James Webb, Caleb Kemere.

**Supervision:** Caleb Kemere, Matthew McGinley.

**Writing – original draft:** James Webb, Caleb Kemere, Matthew McGinley.

**Writing – review & editing:** James Webb, Paul Steffan, Benjamin Y Hayden, Daeyeol Lee, Caleb Kemere, Matthew McGinley.

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
