## [Decision Letter · Decision Letter 0]

24 Sep 2024

Dear Dr Webb,

Thank you very much for submitting your manuscript "Hierarchical Bayesian inference during foraging under uncertainty" for consideration at PLOS Computational Biology. As with all papers reviewed by the journal, your manuscript was reviewed by members of the editorial board and by several independent reviewers. The reviewers appreciated the attention to an important topic. Based on the reviews, we are likely to accept this manuscript for publication, providing that you modify the manuscript according to the review recommendations.

Note in particular that while both reviewers agree the paper makes a significant contribution, the context of that contribution could be more clearly explained, specifically in comparison to alternative foraging models. Also both reviewers request clearer justification (or assessment of the effects) of somewhat arbitrary cut-off boundaries for data exclusion.

Sincerely,

Barbara Webb

Academic Editor

PLOS Computational Biology

Lyle Graham

Section Editor

PLOS Computational Biology

Reviewer's Responses to Questions

**Comments to the Authors:**

Reviewer #1: Review is uploaded as an attachment

Reviewer #2: The authors present a study of patch-leaving behavior in mice under environment conditions that vary, deterministically or stochastically, both within- and between-sessions. Consistent with extensive prior evidence in a wide range of organisms and environments, that mice generally follow the qualitative predictions of the ecological optimum, but with quantitative deviations (“overharvesting”). Through a series of regression and model-based analyses, they determine that the quantitative deviations are best explained by a model that assumes mice are continually updating two estimates of reward availability, reflecting local and global statistics, which are then combined in a manner broadly consistent with Bayesian principles. The coarse observations of the freely-moving variants of the task are replicated in a head-fixed version, supporting the feasibility of using this task design for studying neural mechanisms.

Overall this study is a valuable contribution to the literature, consisting of a novel, well-motivated task designs that generate a rich dataset in which rigorous, comprehensive statistical analysis and model-based insights are consistent with recent work in humans and non-human primates (and perhaps consistent with some early indications in rats, referred below). Extending these to a model organism for which more sophisticated neuroscientific tools promise deeper insights into putative evolutionarily-preserved mechanisms. My questions and comments are minor, should not impede publication, and largely focus on (a) examining in more detail the correspondence with recent findings in humans and NHP in related tasks and (b) clarifying some of the methods used.

1. What age were the mice? This is important because recent work suggests that exploration strategies in standard reward tasks and also more simply structured foraging tasks vary noticeably with development stage (in humans, NHP, and mice — see e.g. Johnson & Wilbrecht 2011, Dev Cog Neuro), and also because it would be valuable to understand whether the feasibility of training mice in this task at early developmental stages has been established. I apologize if this was reported somewhere and I missed it—if so, please copy the report into the ‘Animals’ subsection of Methods and Materials.

2. The decision to exclude the tenth percentile of travel time is reasonable, but raises the question of how robust the results are to this somewhat arbitrary thresholding. Can the authors provide evidence, up to and perhaps including if necessary a repeat of the model-based analyses at different values of this threshold? Of course, if the results are in fact sensitive to this thresholding, it would not be a reason to discount the results of the study; rather, it would provide valuable information for future researchers who wish to adopt this task and associated models.

2a. Relatedly, can the authors more finely characterize the behaviors performed during these excluded transits? Grooming is clear, but (as mentioned in the Discussion section) ‘exploring’ may refer to several different kinds of behavior, some of which may be relevant for inferring mechanisms—e.g. deliberation, “vicarious trial and error”-like head swivels, changes of mind, etc. Of course, this may be difficult or infeasible given the large number of transits involved—a broad characterization of a subset of qualitatively distinct behaviors would be sufficient.

2b. Of particular interest is whether these longer travel times might in some cases be consistent with a model in which the mice are attempting to harvest information about potentially multimodal reward distributions, rather than responding to a higher local reward stochasticity alone [see e.g. Harhen & Bornstein 2022, 2023 in humans and for an explicit implementation of this model; see also concordant findings by Garcia, Gupta, Wikenheiser 2023 in rats; Barack et al 2024 in humans and monkeys]. It’s difficult to tell whether the data support this idea, though a motivated inspection of Fig. 4B,C may suggest some multimodality in residence time deviations from the average. Of course, a key difference in this design is the use of perfectly unpredictable transitions between patch types, which may frustrate animals’ attempts to infer distinct reward modes. However, a response to this question might help inform future designs that more closely examine this question.

3. Is there evidence that local and global reward statistics are continually updated in a manner that reflects ongoing learning consistent with an approximate strategy (see e.g. Wilson et al., 2013, Plos Comp Bio), rather than an exact Bayesian approach whose effective ‘learning rate’ may asymptote with experience? Again, either finding would be consistent with the authors’ overall conclusions—a response to this question is primarily useful for guiding future analysis and research designs.

**Have the authors made all data and (if applicable) computational code underlying the findings in their manuscript fully available?**

Reviewer #1: **No: **The authors state that data and code will be made available upon publication

Reviewer #2: **No: **

PLOS authors have the option to publish the peer review history of their article (what does this mean?). If published, this will include your full peer review and any attached files.

Reviewer #1: No

Reviewer #2: No

Figure Files:

Data Requirements:

Reproducibility:

References:

---

## [Decision Letter · Decision Letter 1]

21 Mar 2025

Dear Dr Webb,

We are pleased to inform you that your manuscript 'Foraging animals use dynamic Bayesian updating to model meta-uncertainty of environment representations' has been provisionally accepted for publication in PLOS Computational Biology.

I note also that the data and code for this paper are not, as yet, made publicly available. Please note to align with journal policy it should be made available as supplementary material or in a public repository before publication.

Best regards,

Barbara Webb

Academic Editor

PLOS Computational Biology

Lyle Graham

Section Editor

PLOS Computational Biology

Reviewer's Responses to Questions

**Comments to the Authors:**

Reviewer #1: I appreciate the revisions the authors have made in response to my previous comments (Reviewer #1). The additional discussion added to the Abstract, Introduction, and Discussion sections help to clarify the novelty of the study design / findings and better distinguish them from other foraging studies / models. The increased focus on meta-uncertainty has particularly helped to convey this message. I also think that the Discussion better situates the results within the context of previous research in behavioral ecology while continuing to emphasize the study’s relevance to future applications in neurophysiology.

In addition to the above revisions, the explanation for the use of acoustic cues was helpful for better understanding its utility, and I found the added clarification in the manuscript to be helpful.

Lastly, I appreciate the explanation provided for the differences in outlier detection methods between the freely moving and head-fixed mice.

Overall, I am satisfied with these revisions and I believe this manuscript will make a strong contribution to the field.

Reviewer #2: The authors have satisfactorily responded to all of my comments. I think this is an excellent contribution to the field.

**Have the authors made all data and (if applicable) computational code underlying the findings in their manuscript fully available?**

Reviewer #1: **No: **The authors state that data and code will be made available to reviewers upon request and at publication

Reviewer #2: None

PLOS authors have the option to publish the peer review history of their article (what does this mean?). If published, this will include your full peer review and any attached files.

Reviewer #1: No

Reviewer #2: No

---

## [Editor Report · Acceptance letter]

PCOMPBIOL-D-24-00819R1

Foraging animals use dynamic Bayesian updating to model meta-uncertainty in environment representations

Dear Dr Webb,

I am pleased to inform you that your manuscript has been formally accepted for publication in PLOS Computational Biology. Your manuscript is now with our production department and you will be notified of the publication date in due course.

With kind regards,

Anita Estes
